# communications

# biology

# Atomic force microscopy-single-molecule force spectroscopy unveils GPCR cell surface architecture

Etienne Dague [1,6✉], Véronique Pons [2,6], Alexandre Roland[2], Jean-Marc Azaïs[3], Silvia Arcucci[2], Véronique Lachaize[1], Samuel Velmont[3], Emmanuelle Trevisiol[1,4], Du N'Guyen[2], Jean-Michel Sénard [2,5] & Céline Galés [2✉]

G protein-coupled receptors (GPCRs) form the largest family of cell surface receptors. Despite considerable insights into their pharmacology, the GPCR architecture at the cell surface still remains largely unexplored. Herein, we present the specific unfolding of different GPCRs at the surface of living mammalian cells by atomic force microscopy-based single molecule force spectroscopy (AFM-SMFS). Mathematical analysis of the GPCR unfolding distances at resting state revealed the presence of different receptor populations relying on distinct oligomeric states which are receptor-specific and receptor expression-dependent. Moreover, we show that the oligomer size dictates the receptor spatial organization with nanoclusters of high-order oligomers while lower-order complexes spread over the whole cell surface. Finally, the receptor activity reshapes both the oligomeric populations and their spatial arrangement. These results add an additional level of complexity to the GPCR phar-macology until now considered to arise from a single receptor population at the cell surface.

[1] LAAS-CNRS, Université de Toulouse, CNRS, Toulouse, France. [2] INSERM, UMR 1297, Institut des Maladies Métaboliques et Cardiovasculaires, Université de Toulouse, INSERM, Toulouse, France. [3] Institut de mathématiques, Université de Toulouse, Toulouse, France. [4] TBI, Université de Toulouse, CNRS, INRAE, INSA, Toulouse, France. [5] Service de Pharmacologie Clinique, Centre Hospitalier Universitaire de Toulouse, Toulouse, France. [6] These authors contributed equally: Etienne Dague, Véronique Pons. ✉email: edague@laas.fr; celine.gales@inserm.fr

A fundamental feature of mammalian cells is their plasma membrane, which integrates multiple transmembrane proteins and converts various extracellular stimuli into intracellular signaling to generate a cellular output and an ensuing physiological response. Among them, G protein-coupled receptors (GPCRs), characterized by seven transmembrane domains, constitute the largest family of cell surface receptors and the most prominent drug targets in pathophysiology[1]. To date, the development of GPCR drugs has relied essentially on the assumption of the existence of a single population of receptor at the cell surface for a given GPCR, i.e., only one receptor target. However, recent fluorescent single-molecule imaging studies based on total internal reflection fluorescence (TIRF) and super-resolution structured illumination microscopy (SIM) indicated that different populations of a given GPCR, relying on different oligomerization states, coexisted at the surface of living cells[2–4]. The behavior of GPCR ligands toward these different receptor populations remains essentially unknown. Hence, there is an urgent need to diversify high-resolution methodologies so to ascertain the native organization of GPCRs at the cell surface. Among them, single-molecule force spectroscopy by atomic force microscopy (AFM-SMFS) has been used in a dynamic mode to quantify the energy landscape of the folding of $\beta_2$-adrenergic receptor ($\beta_2$-AR) after purification and reconstitution into liposomes[5]. However, the force-distance (FD) curves encoding the unfolding pathway of the GPCR will greatly depend on both the chemistry and the physics of the plasma membrane of the cell but also on the GPCR interactome in this membrane. To date, SMFS-analysis of GPCRs in their native environment is still lacking, mainly owing to a majority of non-specific adhesions of the AFM-tip to the cell surfaces. Here, we report for the first time the use of AFM-SMFS to specifically quantify and spatially map GPCRs through their unfolding at the surface of living mammalian cells. An optimized mathematical analysis of the results revealed the unfolding of mainly GPCR oligomers and a tight connection between GPCR activity and its oligomeric architecture at the cell surface.

## Results

**SMFS-based GPCR unfolding at the living cell surface.** SMFS experiments were conducted on adherent living WTT-CHO mammalian cells[6] transfected with an N-terminal (extracellular) HA-tagged GPCR-encoding vector. To specifically probe the HA-GPCR at the cell surface, we took advantage of an AFM tip functionalized with anti-HA antibodies that we previously characterized and validated for it specificity both in vitro and in vivo in WTT-CHO cells to ensure the sensing of highly specific anti-HA/HA-GPCR adhesive events[7,8]. Briefly, the anti-HA antibody was covalently grafted on the AFM tip previously functionalized by aldehyde-dendrimers (Fig. 1a). The resulting anti-HA-dendritip was then approached and retracted from the cell surface at constant velocity with a 0.5 nN maximal applied force and a contact time of 200 ms to optimize the probability of a single interaction between the grafted HA antibody and the HA-GPCR at the cell surface as previously determined[8] (Fig. 1b). Serial approach-withdrawal cycles using the anti-HA-dendritip were conducted while scanning a $3 \times 3$ $\mu m^2$ area at the top of the cell (more precisely the cell surface stretched between the nucleus and the Petri dish), thus allowing the recording of a raster of $16 \times 16$ force-distance (FD) curves with a recording rate of ~2 s/FD curve, i.e. ~10 min/256 FD curves (Fig. 1c and Supplementary Fig. 1). On each retraction force curve (recording rate of ~500 ms; Supplementary Fig. 1), we then manually measured the distance between the contact point and the point of lowest force (Fig. 1d) and proceeded to their mathematical analysis. These unfolded

lengths most likely reflected the length of the HA-receptor and its putative associated proteins unfolded from the membrane at the maximum physically possible extension before rupture of the tip-cell contact (i.e. the anti-HA/HA-GPCR binding) in these specific experimental conditions. Several criteria for FD curve inclusion-exclusion have been established (see Methods; Supplementary Fig. 2), and based on the mean force measured in our previous in vitro characterization of the anti-HA/HA interaction ($61.7 \pm 18.9$ pN)[8], only FD curves displaying forces >40 pN were taken into account and considered as specific anti-HA/HA-GPCR adhesive events. For all experiments, the distribution of the specific unfolding events (adhesive events) was fitted by a mixture of Gaussian distribution according to a method of model selection by penalization likelihood (Bayesian Information Criterion-BIC). In contrast to other Gaussian analyses classically coupled to a defined software with non-accessible and protected algorithms, it provides a customizable-method based on selection criteria (see Methods). Finally, we also resolved the spatial arrangement of both the adhesive events (specific receptor unfoldings) as well as the non-adhesive events (lack of adhesion between the functionalized tip and the cell surface indicative of the absence of HA-GPCR expression) picked up on the 9 $\mu m^2$ scanned area, thus allowing us to map the receptor organization at the cell surface.

**Unfolding GPCRs at steady-state.** To warrant single-molecule interaction between the tip and the cell surface, we first conducted SMFS experiments on WTT-CHO cells expressing low levels of the prototypical family A GPCR, HA-$\beta_2$-adrenergic receptor (HA-$\beta_2$-AR$^{High}$, because of the highest expression of this receptor used in these experiments), as indicated by ELISA-based cell surface detection of the receptor demonstrating an expression level close to the background threshold detected in control WTT-CHO cells in the absence of HA-$\beta_2$-AR expression (Supplementary Fig. 3, **HA-$\beta_2$-AR$^{High}$**). Despite its low expression level, the HA-$\beta_2$-AR$^{High}$ retained its cognate activity on the Gs/adenylyl cyclase pathway to promote cAMP production following iso-proterenol (ISO) agonist-stimulation (Fig. 2a). Under these experimental conditions, in agreement with what we have previously shown[8], we demonstrated the high specificity of the force-extension traces with regard to the detection of HA-$\beta_2$-AR at the surface of WTT-CHO cells, with no more than 18% of adhesive events recorded in control WTT-CHO cells lacking HA-receptor expression (negative control) (Fig. 3a), indicative of non-specific adhesive events, by opposition to 44% of adhesive events in HA-$\beta_2$-AR-expressing cells (Fig. 3b). These 18% adhesive events are not surprising and most likely represent the non-specific adhesion of the anti-HA antibody at the cell surface that is not specific to the AFM technique but more commonly observed in all biological experiments using antibodies that usually introduce blocking steps to prevent such non-specificity. Further supporting the specificity of the anti-HA functionalized tip binding to the HA-receptor expressed at the cell surface, we measured higher rupture forces in HA-$\beta_2$-AR-expressing cells compared to control cells (Fig. 3c, d). The spatial organization of the SMFS-unfolding traces (adhesive events) obtained at the surface of WTT-CHO cells expressing HA-$\beta_2$-AR$^{High}$ or not (control) shows that adhesive and non-adhesive events are homogeneously distributed on the 9 $\mu m^2$ scanned area (Fig. 3e, f). Remarkably, these spatial representations were highly similar in between cells but also experiments, demonstrating the reproducibility of the technique and thus again the specificity of the HA-functionalized tip for HA-GPCR-expressing WTT-CHO cells (Supplementary Fig. 4). As FD curve measurements did not neither differ substantially in between different areas of one cell surface either in terms of the proportion of specific unfolding patterns, non-adhesive events or

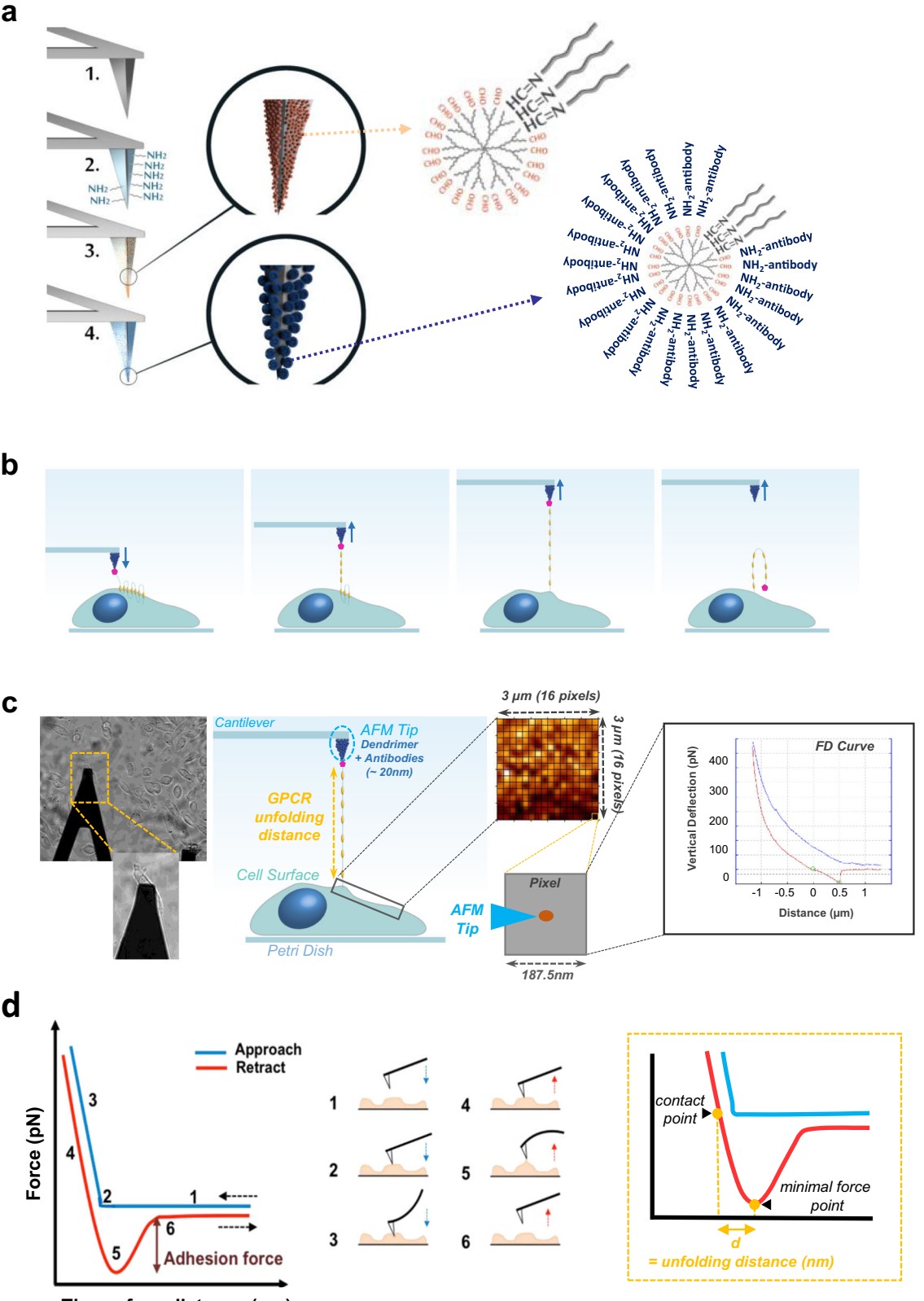

**Fig. 1 SMFS-based GPCR unfolding at the living WTT-CHO cell surface. a** Schematics of the AFM tip functionalization: (1) the silicon nitride AFM tip, and the sequential addition of (2) amine functions, (3) dendrimers, and (4) anti-HA antibodies. **b**, **c**, The functionalized AFM tip is approached to the surface of HA-tagged GPCR-expressing WTT-CHO cells and then retracted until rupture (**b**) on a 3 × 3 μm² area (16 × 16 pixels), resulting in 256 force versus distance curve (FD curve) recordings. **d** Schematics of a representative FD curve showing the different steps of the AFM tip approach (blue) and retraction (red). The unfolding distance is measured between the contact point and the point of minimal force.

**a**

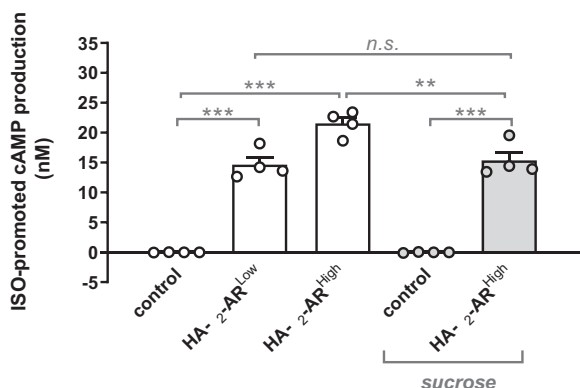

**b**

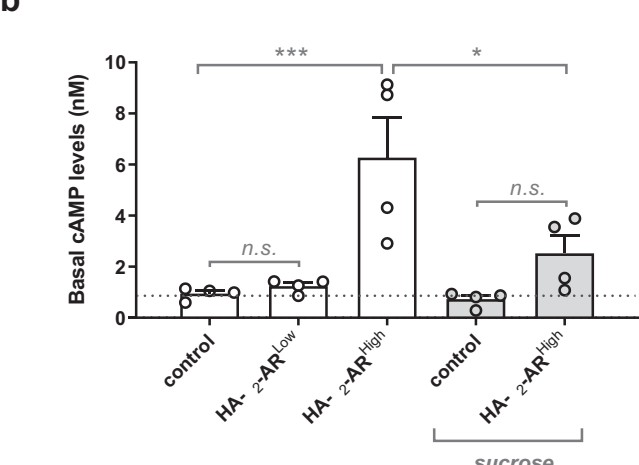

**Fig. 2 Agonist-dependent and agonist-independent HA-$\beta_2$-AR activity in WTT-CHO cells. a** WTT-CHO cells were transfected with different amounts (0.1 or 1 µg) of HA-$\beta_2$-AR-encoding vector (HA-$\beta_2$-AR$^{Low}$ and HA-$\beta_2$-AR$^{High}$ respectively) or the empty vector (control) and cAMP production was quantified following stimulation with 10 µM isoproterenol (ISO) for 30 min (**a**) or at basal state (**b**). When indicated, cells were pretreated with 200 mM sucrose for 1 h to prevent the receptor internalization process. Data represent the mean ± s.e.m. of four independent experiments, each performed in triplicates. The results are expressed as the difference in cAMP level measured in the presence and absence of isoproterenol (isoproterenol-promoted cAMP production) (**a**) or as basal cAMP levels (**b**). The statistical comparison was assessed using one-way ANOVA followed by Sidak's multiple comparisons tests (*$p < 0.05$; **$p < 0.01$; ***$p < 0.001$; n.s. not statistically significant).

spatial arrangement (Supplementary Fig. 5), only one area/cell was analyzed in the subsequent experiments.

Thus, the unfolding distances of HA-$\beta_2$-AR$^{High}$ measured at resting state at the surface of different cells from different independent transfections were pooled and their distribution was analyzed by mixed Gaussian fits according to the BIC-based method. Of note, in contrast to the unfolding of one monomeric $\beta_2$-AR reconstituted in liposomes[5], at constant approach/retraction 0.5 nN velocity, in $\beta_2$-AR-expressing-WTT-CHO cells, we were unable to pinpoint the unfolding force-extension pattern showing seven successive ruptures characteristic of the seven transmembrane domains of this protein using the WLC model fitting, most likely because of the complexity of the interplay

between the GPCR and the native membrane lipids. As shown in Fig. 4a, the Gaussian distribution was better fitted to four populations. Remarkably, while the first population mean (179 nm) correlated with the theoretical length of the HA-$\beta_2$-AR protomer (169 nm), the mean of the larger populations closely related to multiples of two of the theoretical length of the HA-$\beta_2$-AR, thus highly suggesting the main unfolding of receptor oligomeric states. Thus, we found the predominant unfolding of HA-$\beta_2$-AR dimers (350 nm, 39%) and tetramers (669 nm, 32%), as well as receptor monomers (179 nm, 16%) and higher-order oligomers (1262 nm, 13%) but to a lower extent (Fig. 4a and Supplementary Table 1). This assumption of the unfolding of GPCR oligomers by SMFS was confirmed by considering the unfolding of HA-metabotropic glutamate receptor 3 (HA-mGlu3-R), a family C GPCR with approximately twice the theoretical length of HA-$\beta_2$-AR. Accordingly, the distribution of the unfolding distances from HA-mGlu3-R-expressing WTT-CHO cells revealed main Gaussian populations with mean value upward shifts compared to those of HA-$\beta_2$-AR$^{High}$, that were highly related to multiples of the HA-mGlu3-R theoretical length (355 nm) (Fig. 4b, and Supplementary Table 1). It is worth noting that, compared to the HA-$\beta_2$-AR$^{High}$, each main Gaussian mean obtained with the unfolding distances of HA-mGlu3-R was always slightly greater than the theoretical receptor length (see **Discussion**). Although this could most likely originate from a greater standard deviation of the Gaussians, we cannot exclude distinct protomer interconnections within HA-$\beta_2$-AR and HA-mGlu3-R homo-oligomers that could differently impact on their unfolding from the plasma membrane.

To better confirm the unfolding of different oligomeric populations of GPCRs, we performed similar experiments but using the HA-µ-Opioid receptor (HA-µOR) that was recently described to exist essentially as a monomer at the surface of living cells at resting state[9]. In WTT-CHO cells, we probed very high levels of adhesive events (92%) (Supplementary Fig. 6a) similarly to that obtained for the HA-mGlu3-R (Fig. 4j), thus contrasting with the results obtained with the control cells (Fig. 3a) and comforting the specificity of the technique toward HA-GPCR-expressing cells. However, contrary to the TIRF study[9] using very low µOR levels, in WTT-CHO cells expressing low levels of HA-µOR, while we detected a Gaussian peak (188 nm) close to the theoretical length of the receptor monomer (163 nm), the main Gaussian peaks were detected for a population close to a HA-µOR dimer (398 nm) and high-order oligomers (763 nm and 1281 nm) (Supplementary Fig. 6b, c). This result was confirmed by co-immunoprecipitation/Western-blot analysis of the HA-µOR in WTT-CHO cells. Indeed, GPCR promoters are linked together through hydrophobic interactions with their transmembrane domains, thereby conferring a resistance to SDS denaturation and allowing the detection of GPCR oligomers by Western-blot analysis[10]. Thus, Western-blot analyses of the HA-µOR revealed that the monomeric receptor co-exists with higher-oligomeric states (Supplementary Fig. 6d). However, it is noteworthy that even at the lowest expression level of the HA-µOR, the high-order oligomers predominate over the monomer, thus highly supporting the Gaussian analysis of the unfolding lengths obtained by AFM-SMFS (Supplementary Fig. 6b) that depicted a weak Gaussian peak close to the monomer (188 nm) while the other peaks related to the unfolding lengths of dimers and high-order oligomers largely prevailed. Similar co-immunoprecipitation/Western-blot analysis of WTT-CHO cells expressing the HA-$\beta_2$-AR$^{High}$ or the HA-mGlu3-R also confirmed the distribution of the unfolding traces obtained with these two receptors in AFM-SMFS (Fig. 4a, b). Thus, we identified multiple HA-$\beta_2$-AR monomers (arising from different receptor glycosylations, phosphorylations...) and several high-order oligomers but to a

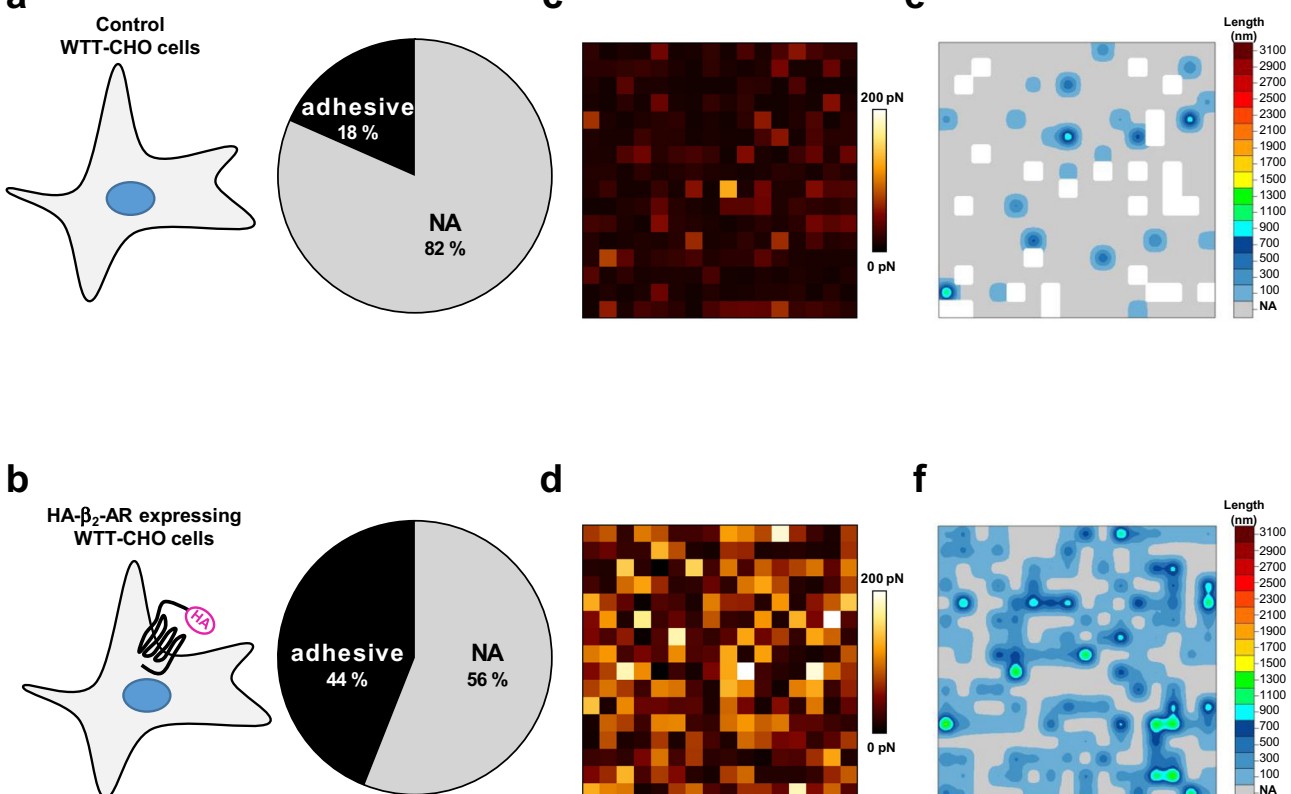

**Fig. 3 Force-distance recorded by SMFS in HA-GPCR-non-expressing and expressing WTT-CHO cells.** SMFS experiments using the anti-HA-functionalized AFM tip were conducted on WTT-CHO cells transiently transfected with (**a**, **c**, **e**) the empty vector as a negative control of specificity for HA-GPCR detection or (**b**, **d**, **f**) with 1 μg of HA-$\beta_2$-AR-encoding vector (HA-$\beta_2$-AR$^{High}$). **a**, **b** The proportions of adhesive and non-adhesive (NA) events detected in the SMFS experiments are shown in the pie charts. **c**, **d** Adhesion forces and (**e**, **f**) unfolding distances resulting from SMFS performed on one representative cell area (3 × 3 μm²) were depicted as a spatial map. At least nine cells from three independent experiments were analyzed.

lower extent (Supplementary Fig. 7a), correlating with the main unfolding distances obtained with this receptor (Fig. 4a), while high proportions of both monomers and high-order populations were depicted for the HA-mGlu3-R (Supplementary Fig. 7b) in agreement with the main Gaussian peaks obtained with the unfolding distances associated with this receptor (Fig. 4b).

Since GPCR oligomerization is a dynamic process that is cell- and expression level-dependent, most likely accounting for the different results in the literature for a given receptor, and that nobody really knows in biology for an effective control for any transmembrane protein monomer or dimer, to definitively prove that the AFM-SMFS unfolding traces mainly reflect the unfolding of HA-tagged receptor oligomers expressed at the cell surface, we designed an artificial positive control of a GPCR dimer consisting in a fusion protein connecting two protomers of $\beta_2$-AR (HA-2x$\beta_2$-AR chimera) with a theoretical length of 315 nm that is close to the length of one mGlu3-R protomer (355 nm). As shown in Fig. 4c, the unfolding distances obtained with the HA-2x$\beta_2$-AR chimera indicate main rupture lengths with Gaussian peak perfectly fitting with the theoretical length of a chimera monomer (350 nm), a dimer (610 nm), a trimer (1004 nm) and a pentamer (1611 nm), clearly reinforcing the notion that the SMFS method reliably detects GPCR oligomers. Of note, in WTT-CHO cells expressing the HA-2x$\beta_2$-AR chimera, we also found a Gaussian at 63 nm close to that of found at 77 nm with the mGlu3-R and similarly representing less than 5% of all the adhesive events. Such a Gaussian was never observed with the other GPCRs with lower theoretical lengths. Whether this peculiar feature of long GPCRs reflects abortive ruptures due to the insertion of these

long transmembrane proteins in specific nanodomains of the plasma membrane difficult to unfold is unknown. Of note, despite rupture forces obtained in WTT-CHO cells expressing HA-$\beta_2$-AR$^{High}$, HA-mGlu3-R or HA-2x$\beta_2$-AR chimera clearly differ from that obtained in control cells lacking expression of HA-tagged protein (Supplementary Fig. 8), we didn't find any correlation between the force and the unfolded distance in these three experimental conditions.

In summary, altogether, the AFM-SMFS results indicated that HA-$\beta_2$-AR and HA-mGlu3-R coexist at the cell surface of WTT-CHO cells as different receptor populations. The presence of multiple HA-$\beta_2$-AR or HA-mGlu3-R complexes at the surface of living cells at resting state, from monomers and dimers to higher-order oligomers, is also consistent with recent findings using other high-resolution methods based on single-molecule imaging[3,4,11,12].

We next resolved the spatial organization of the SMFS-unfolding traces obtained on the 9 μm² area at the surface of HA-$\beta_2$-AR$^{High}$- and HA-mGlu3-R-expressing WTT-CHO cells. Strikingly, we found in both cases a striking and specific architecture of low-order versus high-order oligomers (Fig. 4e, f and Supplementary Fig. 9). Indeed, while short unfolding lengths (low-order complexes) were distributed homogeneously on the overall cell surface, in contrast, long unfolding lengths, indicative of high-order GPCR oligomers, were prone to organize into clusters. Another intriguing key feature of the spatial map of HA-$\beta_2$-AR$^{High}$ unfolding relied on its high degree (56%) of non-adhesive events (Fig. 4i) compared with HA-mGlu3-R (4%) (Fig. 4j) or HA-μOR (8%) (Supplementary Fig. 6a). Accordingly,

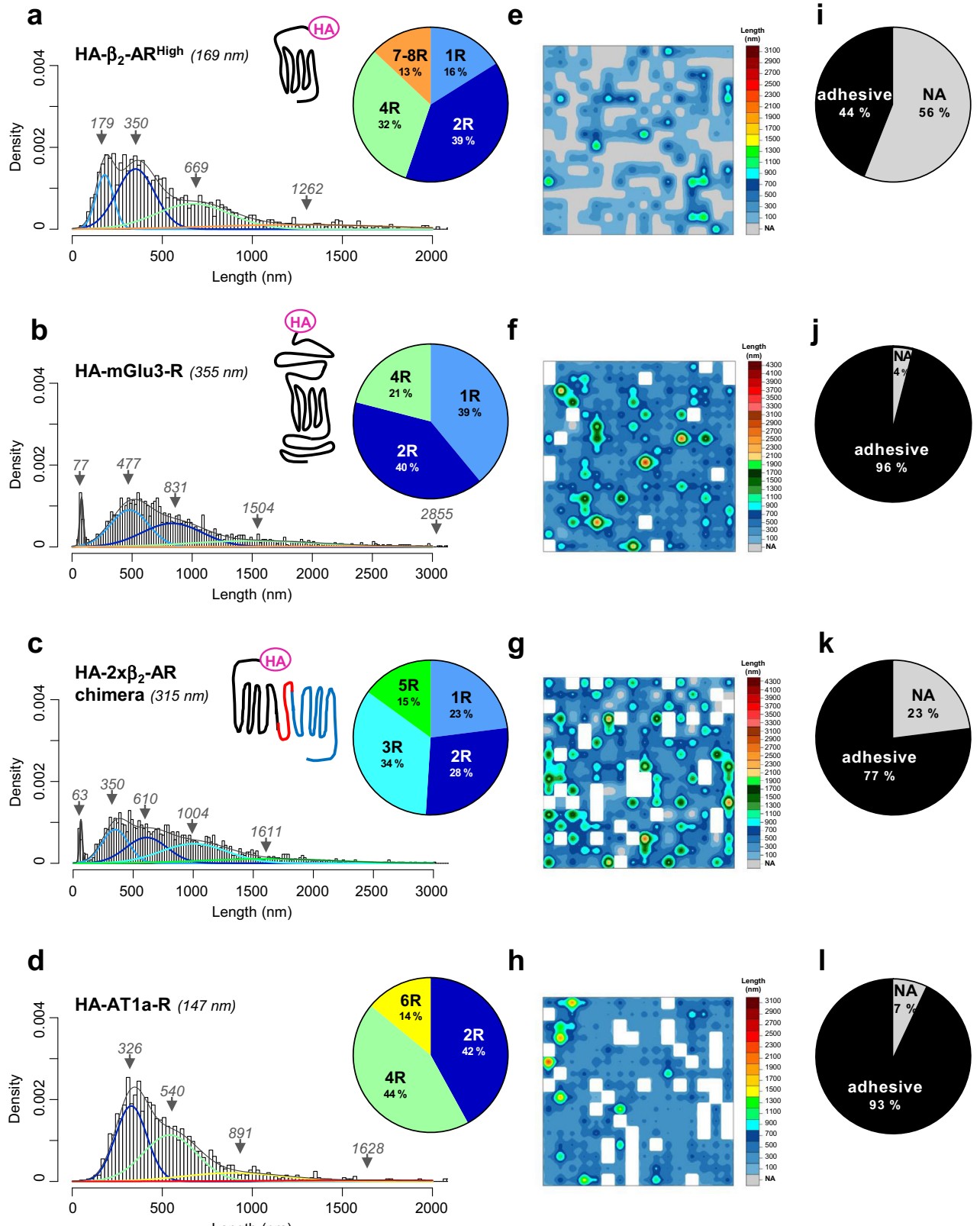

again agreeing with a peculiar feature of HA-$\beta_2$-AR$^{High}$-expressing cells showing high levels of non-adhesive events at the cell surface, the unfolding map of another closely related family A GPCR, the HA-angiotensin II subtype 1a receptor (HA-AT1a-R) with a 147 nm theoretical length, resemble that of HA-mGlu3-R or HA-μOR with a low percentage (7%) of non-

adhesive events (Fig. 4l). Notably, the unfolding of HA-AT1a-R at basal state similarly revealed different oligomeric receptor populations (Fig. 4d). Likewise, the spatial map of HA-AT1a-R unfolding confirmed a specific assembly of high-order receptors as nanoclusters while low-order complexes spread homogeneously on the overall cell surface (Fig. 4h). Taken together, these

**Fig. 4 Mapping GPCR oligomerization and spatial organization using FD-based SMFS at the basal state.** SMFS experiments were conducted at the basal state on WTT-CHO cells transiently transfected with 1 μg of vector encoding (**a**, **e**, **i**) HA-$\beta_2$-AR (HA-$\beta_2$-AR$^{High}$), (**b**, **f**, **j**) HA-mGlu3-R, (**c**, **g**, **k**) HA-2x$\beta_2$-AR chimera or (**d**, **h**, **l**) HA-AT1a-R. Unfolding distances were analyzed by fitting with a Gaussian mixture according to the BIC-based method. Each Gaussian population was assigned to a GPCR oligomeric state (*1 receptor/monomer=1 R; 2 receptors/dimer=2 R; 3 receptors/trimer=3 R...*) based on their theoretical length, and Gaussian weight is presented in the pie charts (**a**, **b**, **c**, **d**). The 256 unfolding distances resulting from one representative cell area (3 × 3 μm$^2$) are depicted as a spatial map (**e**, **f**, **g**, **h**). The proportion of adhesive and non-adhesive (NA) events are shown in the pie charts (**i**, **j**, **k**, **l**). For each condition, at least nine cells from at least three independent experiments were analyzed. Note that **e**, **i** are similar to Fig. 3f, b to facilitate comparisons.

results indicated that GPCRs exist in the basal state as different oligomeric populations that are qualitatively and quantitatively dependent on the nature and the expression level of the receptor.

The specificity of HA-$\beta_2$-AR did not rely on a lower expression of this receptor at the cell surface but more likely to an idiosyncrasy of $\beta_2$-AR. Indeed, further lowering the expression of HA-$\beta_2$-AR (HA-$\beta_2$-AR$^{Low}$) should be theoretically correlated with an increase of non-adhesive events due to lower cell surface expression of the receptor as confirmed with undetectable HA-$\beta_2$-AR$^{Low}$ by cell surface ELISA compared to HA-$\beta_2$-AR$^{High}$ conditions (Supplementary Fig. 3). However, and contrary to the assumption, lowering the expression of the HA-$\beta_2$-AR instead led to a substantial decrease in non-adhesive events (Fig. 5h) compared with those measured at a higher HA-$\beta_2$-AR expression level (HA-$\beta_2$-AR$^{High}$) (Fig. 5g), thus indicating a new receptor organization at the cell surface that is expression-dependent. This result was corroborated by a significant change in the oligomerization state of the receptor with the extinction of $\beta_2$-AR dimer, tetramer, and octamer populations together with an important loss of receptor monomers (16 versus 9%), conversely to the formation of a major population of trimers (81%) and a minor population of new high-order oligomers consisting of five-six receptors (Fig. 5b and Supplementary Table 1). Interestingly, at the spatial level (Fig. 5e), this loss of $\beta_2$-AR oligomers correlated with the loss of clusters and a reshaping of the low-order receptor complexes spread over the whole surface.

Because $\beta_2$-AR is a prototypical GPCR harboring high constitutive activity (CA)[13], i.e., agonist-independent activity, we questioned about a potential role of the CA in the particular spatial profiling of the HA-$\beta_2$-AR$^{High}$ architecture at the WTT-CHO cell surface. More specifically, we wondered whether receptor CA in the internalization process could significantly deplete the receptor from the cell surface, thus accounting for the high level of non-adhesive traces depicted in the SMFS experiments (Fig. 5g). We first validated that the HA-$\beta_2$-AR$^{High}$ exhibited CA in WTT-CHO cells. However, because the HA-$\beta_2$-AR expression level at the cell surface used in this study was close to the background (Supplementary Fig. 3 - HA-$\beta_2$-AR$^{High}$), we could not finely examine the CA on receptor internalization by quantifying its cell surface expression level. Hence, we analyzed the CA of the HA-$\beta_2$-AR$^{High}$ by examining its cognate Gs-cAMP signaling known to occur in endocytic vesicles[14,15]. As shown in Fig. 2b, in the absence of agonist stimulation, the expression of higher levels of HA-$\beta_2$-AR (HA-$\beta_2$-AR$^{High}$) promoted a significant increase in basal cAMP production compared to WTT-CHO control cells, while CA was lost when decreasing the expression of the receptor (HA-$\beta_2$-AR$^{Low}$), thus agreeing with the well-known receptor density dependency of GPCR CA[16]. Hence, HA-$\beta_2$-AR$^{High}$ displayed CA on the Gs-cAMP pathway only at higher expression levels in WTT-CHO cells. Since most of the overall cAMP response promoted by the $\beta_2$-AR has been previously reported to originate from the internalized receptor pool[14,15], we next evaluated the contribution of the internalization process to the CA of HA-$\beta_2$-AR$^{High}$ on the Gs-cAMP pathway. Consistently, preventing the $\beta_2$-AR internalization

process by sucrose pretreatment[17] significantly blocked basal cAMP production in HA-$\beta_2$-AR$^{High}$-expressing cells (Fig. 2b), thus indirectly indicating constitutive internalization of HA-$\beta_2$-AR$^{High}$. To further explore a putative constitutive $\beta_2$-AR internalization that would quantitatively impact the non-adhesive events at the cell surface quantified in the AFM-SMFS experiments, we either adjust the receptor density, which is correlated with the spontaneous activity of GPCRs[16] as we have confirmed previously in the WTT-CHO cells (Fig. 2b) or we directly blocked the $\beta_2$-AR internalization process[17]. Thus, lowering the HA-$\beta_2$-AR expression level (HA-$\beta_2$-AR$^{Low}$) or blocking receptor internalization through sucrose pretreatment, both led to a significant loss of the non-adhesive events (Fig. 5h, i) when compared to a higher HA-$\beta_2$-AR expression (HA-$\beta_2$-AR$^{High}$) (Fig. 5g), thus demonstrating that non-adhesive events originated from agonist-independent internalization of the receptor. Remarkably, both experimental modulations of HA-$\beta_2$-AR CA promoted a similar qualitative and quantitative modification of receptor unfolding (Fig. 5b, c and Supplementary Table 1). In both cases, a striking and common feature was the loss of the unfolding of HA-$\beta_2$-AR dimers, tetramers and the higher-order oligomers (7–8 R), together with the reorganization of the receptor in a prominent trimeric population spread over the whole cell surface (Fig. 5e, f), strongly suggesting that $\beta_2$-AR dimers and multiples of dimers constituted the active form of the receptor. Of note, to more directly link the HA-$\beta_2$-AR$^{High}$ CA to the oligomerization/architecture profiling of the receptor, we also performed AFM-SMFS experiments on HA-$\beta_2$-AR$^{High}$-expressing cells in the presence of the $\beta_2$-AR specific inverse agonist ICI-118,551[18]. However, due to a receptor-independent effect of ICI-118,551 weakening the WTT-CHO cell surface to the AFM-tip contact, we were unable to proceed with these experiments.

**Unfolding GPCRs following long-term agonist stimulation.** Finally, because SMFS experiments do not allow studies of short-term dynamic events, we next examined whether long-term stimulation with agonists could influence the GPCR oligomeric structure and its spatial organization at the cell surface. It is generally accepted that prolonged agonist-stimulation of GPCRs leads to a negative regulation of the receptor activity known as the desensitization process. Interestingly, the stimulation of HA-$\beta_2$-AR$^{High}$- or HA-AT1a-R-expressing WTT-CHO cells for 20 min with 10 μM isoproterenol (ISO) or angiotensin II (AngII), both led to a general and significant shift and collapse of the high-order oligomers (>4 receptors) in favor of the formation of receptor dimers (Fig. 6a–d) together with a new spatial organization of the receptors (Fig. 6e–h). It is interesting to note that despite the two agonists acting on different basal cell surface organization of their respective GPCR targets (Fig. 6a, c), they ultimately resulted in the long-term in similar reorganization of the receptors at the cell surface both in terms of the proportion of adhesive versus non-adhesive traces (Fig. 6i–l), the amount of receptor oligomers (Fig. 6a–d) or their spatial partitioning at the cell surface (Fig. 6e–h).

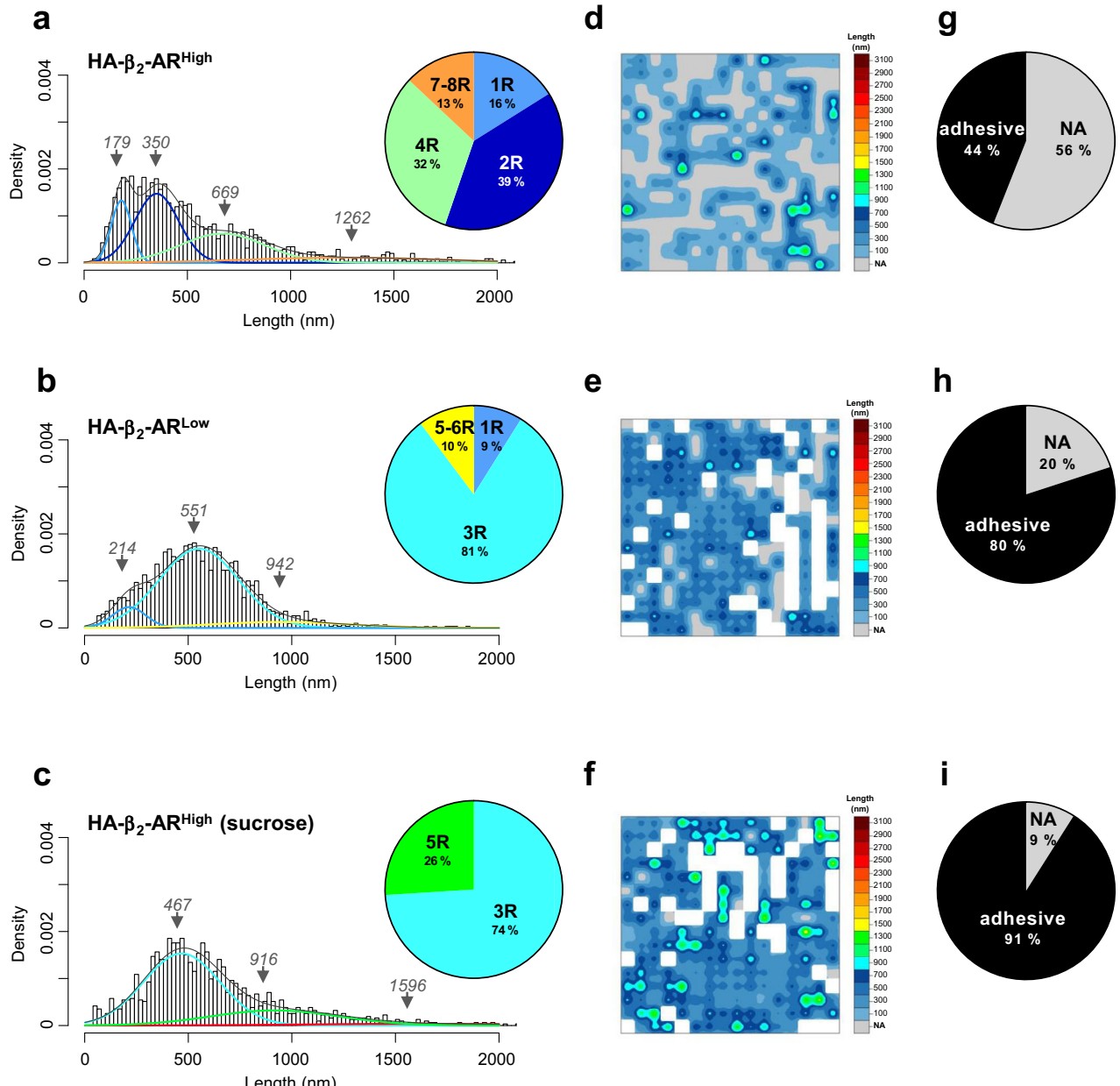

**Fig. 5 Correlating the GPCR constitutive activity with their oligomerization and spatial organization using FD-based SMFS.** SMFS experiments were conducted on WTT-CHO cells transiently transfected with 1 μg of vector encoding HA-β$_2$-AR (HA-β$_2$-AR$^{High}$) treated (**c**, **f**, **i**) or not (**a**, **d**, **g**) with 200 mM sucrose for 1 h or with 0.1 μg of HA-β$_2$-AR-encoding vector (HA-β$_2$-AR$^{Low}$) (**b**, **e**, **h**). Unfolding distances were analyzed by fitting with a Gaussian mixture according to the BIC-based method. Each Gaussian population was assigned to a GPCR oligomeric state (*1 receptor/monomer=1 R; 2 receptors/dimer=2 R; 3 receptors/trimer=3 R...*) based on their theoretical length, and Gaussian weight is presented in the pie charts (**a**, **b**, **c**). The 256 unfolding distances resulting from one representative cell area (3 × 3 μm$^2$) are depicted as a spatial map (**d**, **e**, **f**). The proportions of adhesive and non-adhesive (NA) events are shown in the pie charts (**g**, **h**, **i**). For each condition, at least nine cells from at least three independent experiments were analyzed. Note that **a**, **d**, **g** are similar to Fig. 4a, e, i to facilitate comparisons.

These results demonstrated that in contrast to the basal state, long-term agonist stimulation promoted a common oligomeric rearrangement and a spatial reorganization of the GPCRs at the cell surface, at least for two highly different GPCRs, namely, β$_2$-AR and AT1a-R.

## Discussion

AFM-SMFS allows force probing of the surfaces of living cells at molecular resolution and provides information about cell surface receptors[19–21]. To date, in the GPCR field, AFM-SMFS has been essentially dedicated to the unfolding of GPCRs over a range of different loading rates (dynamic mode) to appreciate the energy landscape of their structural state as for β$_2$-AR in liposomes[5] or rhodopsin in native membranes[22] and more recently to study the ligand binding properties of protease-activated receptor-1 in proteoliposomes[22,23]. In this study, we present for the first time the use of AFM-SMFS in the force-volume mode to specifically pull out GPCRs from the surface of living cells in their native cellular environment, allowing us to extract information about their architectural arrangement and to map their spatial organization at steady-state and their reshaping upon receptor activation.

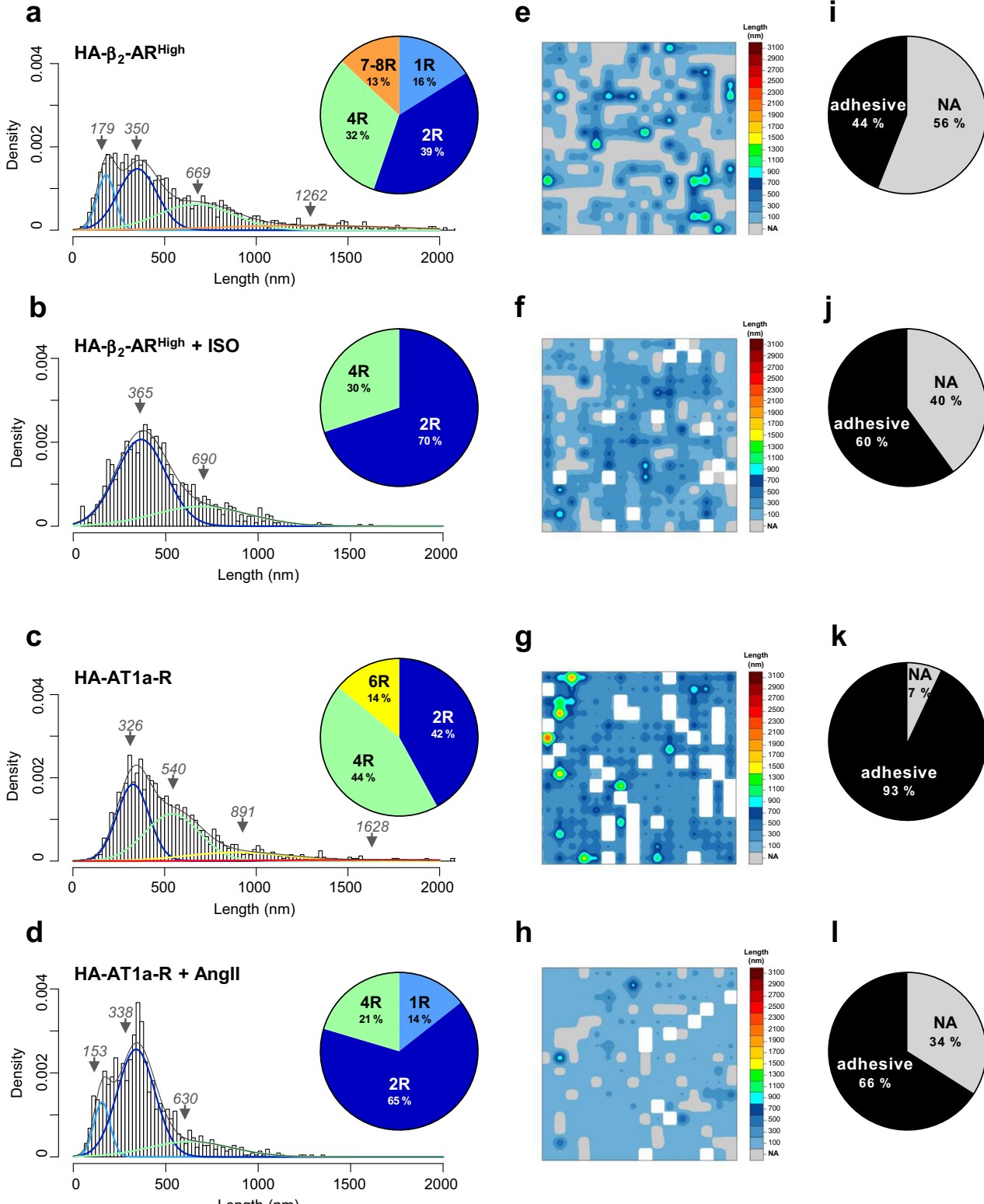

**Fig. 6 Mapping agonist stimulation on GPCR oligomerization and their spatial organization using FD-based SMFS.** SMFS experiments were conducted on WTT-CHO cells transiently transfected with 1 μg of vector encoding HA-β2-AR (HA-β2-AR$^{High}$) or HA-AT1a-R and treated or not (**a**, **e**, **i**, **c**, **g**, **k**) with 10 μM isoproterenol (ISO) (**b**, **f**, **j**) or angiotensin II (AngII) (**d**, **h**, **l**) respectively for 20 min. Unfolding distances were analyzed by fitting with a Gaussian mixture according to the BIC-based method. Each Gaussian population was assigned to a GPCR oligomeric state (*1 receptor/monomer=1 R; 2 receptors/dimer=2 R; 3 receptors/trimer=3 R…*) based on their theoretical length, and Gaussian weight is presented in the pie charts (**a**, **b**, **c**, **d**). The 256 unfolding distances resulting from one representative cell area (3 × 3 μm²) are depicted as a spatial map (**e**, **f**, **g**, **h**). The proportions of adhesive and non-adhesive (NA) events are shown in the pie charts (**i**, **j**, **k**, **l**). For each condition, at least nine cells from at least three independent experiments were analyzed. Note that **a**, **e**, **i**, **c**, **g**, **k** are similar to Fig. 4a, e, i, d, h, l or Fig. 5a, d, g to facilitate comparisons.

In this study, the assumption that major unfolding events depicted by AFM-SMFS reflected the oligomerization states of GPCRs relies on the theoretical length of each HA-tagged receptor amino acid sequence that we compared to the Gaussian distribution mean of the unfolding distances estimated from the FD curves. This assumption was clearly confirmed by: i/ the unfolding of a positive control for GPCR dimer (HA-2x$\beta_2$-AR chimera) consisting in a fusion between two $\beta_2$-AR protomers for which the main unfolding distances measured by Gaussian analysis perfectly fitted with one or multiple of the theoretical length of this fusion protein, and ii/ another biochemical technique of co-immunoprecipitation/Western-blot performed under the same experimental conditions as those for the AFM-SMFS, and allowing the detection of different monomer/oligomer GPCR populations, in agreement with the main unfolding distances measured from the AFM-SMFS records. Although the unfolding distances were assessed manually by independent blind experimenters and based on different inclusion/exclusion criteria that were a priori defined, the approach is reliable and robust as indicated by the perfect linear correlation between the standard deviation and the mean of each Gaussian population obtained for the different experimental conditions (Supplementary Fig. 10). Thus, as expected, we found a better statistical evaluation of the short distance means, by far the best represented, compared to long distances that tend to be much less abundant. This could, in part, account for the discrepancies obtained between the pattern of HA-$\beta_2$-AR$^{High}$ unfolding with the major unfolding of short distances compared with HA-mGlu3-R (Fig. 4a, b). Indeed, HA-mGlu3-R with a theoretical contour length twice that of HA-$\beta_2$-AR was taken as a positive control in this context to ascertain that each Gaussian population mean obtained by GPCR experimental unfolding reflected a specific population of receptor oligomers. Moreover, as a class C GPCR, this receptor represents a prototypical mandatory dimer relying on its disulfide-linked homodimer. However, the analysis of unfolding distances in HA-mGlu3-R-expressing cells identified a Gaussian population accounting for 39% of the total adhesive events with a mean below the theoretical length of a dimer and closer to one protomer length, and thus assigned to a HA-mGlu3-R monomer. In fact, distinct protomer interconnections between HA-$\beta_2$-AR and HA-mGlu3-R could differently influence their unfolding from the plasma membrane. Different oligomer interfaces have been suggested for different homodimers in reconstituted vesicles[24]. In this context, the unfolding pattern of HA-$\beta_2$-AR by AFM-SMFS could indicate a linear or "zigzagged" organization between dimers of dimers to form GPCR homotetramers or homooctamers as proposed for $\mu$OR heterotetramers, while a large transmembrane domain interconnection between two HA-mGlu3-R protomers could most likely agree with high-resolution GPCR crystal structures in which interaction through the transmembrane domains of each protomer predominates[25]. It follows that even the unfolding events of HA-mGlu3-R were classified as monomers, they could reflect the unfolding of a dimer with a tight intertwining between the two protomers. The same is also true for higher-order HA-mGlu3-R unfoldings. Although these results could indicate a different 3D organization of the protomer within the plasma membrane between $\beta_2$-AR and mGlu3-R, a weaker statistical confidence due to larger unfolding distances of HA-mGlu3-R (Supplementary Fig. 10) could also contribute to an approximation of the nature of the oligomer, thus reaccrediting the existence of mGlu3-R nanoclusters containing mainly monomers and dimers (Fig. 4b, f). Accordingly, although class C GPCRs have been essentially described as minimal dimers at the cell surface, super-resolution imaging in native neurons recently revealed the existence of nanoclusters of mGlu4-R, closely related to mGlu3-R, containing mainly one to

two subunits of the receptor[12]. The existence of high proportions of mGlu3-R monomers in WTT-CHO cells detected by AFM-SMFS was further confirmed in our study through the use of another technique, i.e. co-immunoprecipitation/Western-blot experiments (Supplementary Fig. 7b). This result reinforces the notion that it is impossible to assign a unique oligomeric state signature to a GPCR since it will be highly cell-dependent (due to different lipids in the plasma membrane[26], different expression levels[2] and activation states of the receptor[9]…). Accordingly, while recently the $\mu$OR was assigned essentially, but not exclusively, to a monomeric state, in our WTT-CHO specific cell system and experimental conditions, we identified a mixture of both $\mu$OR monomers and high-order oligomers in AFM-SMFS that we also confirmed by co-immunoprecipitation/Western-blot experiments. Alternatively, because the unfolding of the two GPCRs in our study was performed in a non-dynamic mode at constant retraction velocity of the AFM cantilever and the two receptors display different biochemical/biological properties supported by a main difference in size, we cannot exclude the possibility that it could differentially influence protein unfolding with, for instance, a hasty rupture of the HA-mGlu3-R due to the largest dimer stabilization of the structure inserted within the plasma membrane. In line with this assumption, it was reported by AFM-SMFS that protein oligomerization stabilized the folding of one protein protomer[27] but also that different plasma membrane environments (i.e., cholesterol) can modify the mechanical properties of GPCR unfolding[5,28]. Thus, large complexes of the GABA$_B$ receptor, another class C GPCR, were found to tether to the cortical actin cytoskeleton[2]. Finally, the difference between the theoretical length of HA-mGlu3-R and its unfolding distance on living cells could also arise from oligomeric structures of the receptor-interacting with other specific transmembrane proteins and pulled out together with the receptor since it is well known that GPCRs can interact with a large panel of accessory proteins[29].

How GPCRs organize at the cell surface in living cells and whether their ligands impact on this organization remain an exciting but technically challenging matter of debate in the field. This receptor family has long been viewed as a single monomeric functional unit at the cell surface that picked up the extracellular agonists and converted them into an intracellular signal. However, it is now well established that, like many other transmembrane receptors, GPCRs can assemble into oligomeric complexes at the cell surface[24]. A fundamental question that now remains is the stoichiometry of these oligomeric states and their behavior toward the extracellular ligands. The answer to that question varies greatly depending on the methodology, the GPCR and the cell system, but recent independent studies using high-resolution single-molecule microscopy to track GPCRs (endogenously expressed or overexpressed) and their mobility at the surface of living cells now converge on the coexistence of different receptor oligomeric populations at steady state at the cell surface[2–4,9,11,12]. Since these results arise essentially from the use of a common TIRF microscopy technique or FRET-based techniques, it is thus important to support them using, in parallel, different techniques using similar experimental conditions (similar cell system, similar receptor expression). Thus, divergent results about the oligomerization state of a GPCR between independent studies could be attributed to different techniques requiring different experimental conditions. Thus, TIRF- or FRET-based studies require low receptor expression levels due to the technique, while, on the contrary, in our AFM-SMFS study, the receptor expression level may not be decreased too much as we might lose the detection sensitivity. Moreover, single-molecule imaging techniques (TIRF, FRET-based) do not rely on the use of native receptors but on GPCRs that have been fused to large tags (SNAP tag-182 amino

acids-~20 kDa or GFP-based tags-240 amino acids-~27 kDa) at their N- or C-terminus that could interfere with oligomer complex formation and could also affect the diffusion rates in between different receptor populations. By contrast, the AFM-SMFS used GPCRs fused at their N-terminus with an HA-Tag consisting in a 9 amino acid sequence (1 kDa) that didn't interfere with receptor expression and activity. For instance, all these differences could account for the discrepancies on the μOR oligomerization state obtained with our AFM-SMFS method or western-blots in WTT-CHO cells and depicting both μOR monomers and high-order oligomers, versus those obtained in different cells with the TIRF or FRET-based techniques[9,30] and measuring most exclusively monomers.

Here, we exploited SMFS by AFM to provide information about their architectural and spatial organization at the cell surface and, in agreement with the different TIRF-based studies, our results demonstrate that i/ different GPCR populations coexist at the cell surface at steady-state, ii/ these receptor populations rely on the oligomeric architecture of the receptor, iii/ the GPCR architecture is receptor specific and dependent on the receptor expression level, iv/ the GPCR spatial organization is dependent on the size of the oligomeric population with higher-order oligomers tending to form nanoclusters, and v/ GPCR activity (ligand-induced or spontaneous constitutive activity) modifies both the architecture and the spatial organization of the receptor. It is worth noting that contrary to TIRF microscopy, AFM-SMFS cannot provide information about the dynamics of the receptors at the cell surface. Hence, results obtained in this study relative to the receptor oligomeric states are indicative of the most stable populations at a given time point, considering both the interaction strength between protomers and/or their interaction speed since, for instance, rapid interconversion between dimer-monomer seems to be a general property of class A GPCRs[31]. This is important to consider when interpreting the AFM-SMFS data at the resting state and the existence of the monomeric and dimeric organization of GPCRs. Indeed, the monomeric and dimeric states were both detected for the β2-AR and mGlu3-R, while only a dimeric population was observed for the AT1a-R. However, this result does not necessarily indicate a lack of AT1a-R monomers, since it could be assigned to different mobility speeds between receptor protomers in favor of the detection of the dimer.

So far, the stoichiometry of GPCR oligomers has been debated essentially on the existence of monomers or dimers and their relation to the receptor active state. Despite all single-molecule high-resolution TIRF studies argued for the coexistence of a larger proportion of both GPCR monomers and dimers at basal state, they also observed, but to a lesser extent, higher-order receptor oligomers[2,3,9,12], which were related to a higher expression level of the receptor[2]. However, nanoclusters of high-order GPCR oligomers were also observed in vivo at physiological expression levels for GLP1-R in the pancreas[11] or mGlu4-R in parallel fiber active zones in the mouse cerebellum[12], most likely suggesting their physiological relevance. Here, our results confirm the coexistence of a majority of monomers and dimers at the basal state for three different GPCRs, as well as the presence of a non-negligible proportion of tetramers and higher-order complexes even at low levels of receptor expression. Whether receptor protomers interconvert in between these different populations, as demonstrated for monomers and dimers[31], still remains an open question.

Another outstanding result from our study is the relationship between the activity of the GPCRs and their structural reorganization (Figs. 5, 6 and Supplementary Fig. 11) and, more specifically, their propensity to dimerize upon activation, thus corroborating a recent TIRF study on the μOR[9]. Hence, we showed that inactivating the constitutive activity of β2-AR by sucrose treatment or lowering receptor expression both led to a striking loss of receptor dimers. Likewise, specific agonists for β2-AR or AT1a-R both led to a large increase in the proportion of receptor dimers. It follows that although different GPCRs can display specific basal architecture at the cell surface, receptor activation-induced dimerization could be a more general feature of this class of receptors. It should be noted that in addition to the modulation of the dimeric population, the receptor active state also correlates with the loss of higher-order receptor populations (>4 receptors). Finally, another major finding of our AFM-SMFS study is the spatial organization of the different unfolded GPCR populations that do not organize randomly at the cell surface of the WTT-CHO cells at resting state even at the low expression level of the receptors, but rather partition into clusters containing high-order receptor oligomers (>4 receptors), while monomers and dimers are more spread out. While this result again corroborates the existence of distinct GPCR populations in a single cell, it is also intriguing to see that agonists can influence the spatial distribution of these populations. In the future, it would be interesting to perform similar experiments but using a range of GPCR ligands exhibiting different pharmacological efficacies, i.e., biased ligands. In line with this perspective, Shen et al. have already reported the existence of different β2-AR signaling pathways assigned to specific oligomeric populations of the receptor that coexist at the surface of a single cell but segregate into different subcompartments[4].

Overall, our study reveals a previously unappreciated potential of AFM-SMFS to depict, by receptor membrane extraction, the architectural and spatial organization of GPCR at the surface of living cells. The data reveal the coexistence of multiple GPCR populations at the cell surface of a single cell and their specific spatial localization together with their remodeling dependent on the receptor activity, thus supporting recent findings using single-molecule TIRF microscopy. Beyond, these results reinforce the complexity of the GPCR machinery and a step further will be to decipher the relevance of each receptor population at the cell surface regarding the ligand binding and efficacy.

## Methods

**Cell culture and transfection**. WTT-CHO cells[6] were cultured in DMEM Glutamax 4.5 g Glucose (Gibco) supplemented with 10% (v/v) fetal bovine serum (Gibco) and 100 units/mL penicillin and 100 μg/mL streptomycin at 37 °C in a humidified atmosphere at 5% $CO_2$. Transient transfections were performed 24 h after cell seeding into 100-mm Petri dishes using X-tremeGENE™ 9 DNA Transfection Reagent (Merck).

**AFM tip functionalization**. Functionalized tips were produced according to a French patent of the authors (FR 2965624) described in sensors and actuators[7]. Briefly, AFM tips with MLCT AUWH cantilevers (nominal spring constant of 0.01 N/m) (Bruker corporation, San Diego, CA) activated with 100% plasma $O_2$ for 3 min were first incubated for 14 h in ethanolamine 50% (w/v) solution in DMSO, followed by three washes with DMSO and three washes with 99% EtOH. Tips were then functionalized by 6 h incubation with 58 μM GC4-CHO'dendrimer solution in THF followed by three THF washes and three 99% EtOH washes, thus allowing the tips to present CHO functional groups able to covalently link with $NH_2$ functional groups of proteins, in this context, of the anti-HA antibody. The dendritips were then incubated with the anti-HA antibody at a concentration of 0.01 mg/mL (LifeProTein, HA.C5 clone monoclonal antibody) for 1 h, before being used for force spectroscopy experiments.

**AFM-SMFS experiments**. For AFM-SMFS experiments, 24 h after transfection, 150,000 cells were split on 35-mm Petri dishes. The next day, non-confluent cells were stimulated or not for 20 min at 37 °C with the GPCR agonists or incubated for 1 h with 200 mM sucrose solution in a humidified atmosphere at 5% $CO_2$. Cells were then rinsed twice, incubated in DMEM Glutamax 4.5 g Glucose supplemented with 15 mM HEPES (4-(2-hydroxyethyl)-1-piperazineethanesulfonic acid), and placed on the PetriDishHeater (JPK) of the AFM microscope, which maintained the Petri dish at 37 °C throughout the AFM experiment. AFM-SMFS experiments were conducted on the JPK Nanowizard III AFM module mounted on a Zeiss Axiovision inverted microscope. Force spectroscopy (maximum applied force of

0.5 nN, constant speed of 3 µm.s⁻¹, Z length of 3 µm,) was used in the force map mode (16 × 16 FC matrices on a 3 × 3 µm² area). Cantilever sensitivity and spring constant were determined for each cantilever at the end of the experiment to avoid harming the functionalized tip. The spring constant was determined by the thermal noise method[32].

Individual force curves of AFM force maps were analyzed using JPKSPM data processing software. The baseline of the retraction curve was corrected using a linear fit on the last 20% of the retraction curve, allowing the positioning of the contact point. Unfolding distances were then manually measured between the contact point and the point of minimum force. These quantifications were initially validated by several observers on few similar AFM experimental recordings and gave similar results based on the established inclusion/exclusion criteria (Supplementary Fig. 2) (Inclusion: Adhesion forces >40pN; Exclusion: disturbances in the approach curve (Supplementary Fig. 2a), tethers (Supplementary Fig. 2c), unidentifiable baseline). Quantifications of the unfolding distances presented in the manuscript were conducted by two observers.

**Quantification of GPCR oligomeric states by Immunoprecipitation/Western-blotting**. For immunoprecipitation experiments, WTT-CHO cells were transiently transfected with pcDNA3.1 (+) empty vector (control) or vectors encoding HA-GPCRs in similar conditions to those used for AFM-SMFS experiments. Forty-eight hours post-transfection, cells were lysed in a lysis buffer (140 mM NaCl/ 2 mM EDTA/25 mM Tris, pH 7.4/0.5% DDM) supplemented with complete protease and phosphatase inhibitors (Roche). Protein extracts (~2 mg) were immunoprecipitated overnight at 4 °C with Dynabeads™ Protein G (Invitrogen) precoated with an anti-HA antibody (BioLegend, 16B12 Clone) and proceeded to western-blot analysis as previously described[33]. Proteins were detected with the anti-HA primary antibodies (Biolegend, 16B12 Clone) followed by antimouse Horse Radish Peroxidase (HRP)-conjugated secondary antibodies (TrueBlot, eB144Clone, Rockland) using enhanced chemiluminescence detection reagent (RPN2232 Prime, GE Healthcare).

**Quantification of cell surface receptors by ELISA**. WTT-CHO cells were transiently transfected in 100-mm Petri dishes with pcDNA3.1 (+) empty vector (control) or vectors encoding N-terminally HA-tagged receptors. Eighteen hours post-transfection, 120,000 cells/well were then split into 24-well plates precoated with poly-D-lysine. Cells were then fixed (4% paraformaldehyde), saturated (PBS - 1% BSA) and incubated with the primary anti-HA antibody (BioLegend, 16B12 Clone) and then with the HRP-labeled secondary antibody (Sigma, St. Louis, MO, USA). After washing, the cells were incubated for 1 h with the TMB (3,39,5,59-tetramethylbenzidine) HRP substrate. The reaction was stopped with 1 N HCl, and the plates were read at 450 nm in a microplate reader (Tecan Infinite F500).

**cAMP accumulation assay**. Quantification of intracellular cAMP was performed using the HTRF cAMP competitive immunoassay (Cisbio. France) according to the manufacturer's instructions. Briefly, 48 h post-transfection, 5,000 WTT-CHO cells/ well were distributed in a 384-well white microplate (Greiner) in the absence (basal) or in the presence of isoproterenol (ISO, 10 µM) for 30 min at 37 °C and in the presence of 0,5 mM IBMX to prevent phosphodiesterase-mediated cAMP degradation. When indicated, cells were pretreated with 200 mM sucrose for 1 h at 37 °C to prevent receptor internalization. After the addition of Cryptate-labeled cAMP (donor) and anti-cAMP-d2 (acceptor) for 1 h, the specific FRET signals were calculated by the fluorescence ratio of the acceptor and donor emission signal (665/620 nm) collected using a modified Infinite F500 (Tecan Group Ltd). Conversion of the HTRF ratio of each sample into cAMP concentrations was performed on the basis of a standard curve to determine the linear dynamic range of the assay.

**Mixture Gaussian model analysis**. The data were analyzed by a model of a mixture of heteroskedastic Gaussian populations. The analysis was performed by maximum likelihood optimization via the EM algorithm using the standard R routine normalmixEM from the "mixtools" package[34]. Because of practical considerations, the number of subpopulations k was chosen in the interval 3–6. The selection of their size was performed using the Bayesian information criterion (BIC), which is favorable for small sizes. In some cases, two sizes had close BIC values, and the final decision was made by visual agreement between the histogram and the estimated density. There is no theoretical result guaranteeing the convergence of the EM algorithm to the true maximum. To check the convergence and following the standard option of normalmixEM, the algorithm was initiated with a random value of the parameter. This operation, in each case, was repeated independently 50 times, and the likelihood was computed. In each case, we manually checked that the solution with the highest likelihood appeared the most often.

The Gaussian curve means were assigned to monomers or to the different receptor oligomers according to the position of their center and their proximity to the theoretical length of each HA-GPCR. Gaussian curves with weight <5% of the total weight were excluded from this analysis. For the Gaussian-pie chart representations, the percentage for each GPCR oligomeric state was calculated from data in supplementary table 1.

**Spatial analysis of SMFS-unfolding traces**. For each sample of each cell, the data obtained on a 16 × 16 array were analyzed using a Kriging algorithm and a home-made program to smooth the data before performing a contour representation using the R routine "filled.contour". Briefly, after subtraction of the mean to the data, Kriging was achieved with an exponential model ($cov(X(i_1, j_1), X(i_2, j_2)) = e^{-\theta dist}$) where $dist$ is the distance between two locations $(i_1, j_1)$, $(i_2, j_2)$ in $[1, 16]^2$ and a value of θ fixed to 0.043. In this model, Kriging is a multiplication of the matrices of covariance with the inverse of the matrix of variance-covariance. After that, the value of the mean was reintroduced and discarded data were highlighted in white, non-adhesive curves were highlighted in grey and adhesive curves were highlighted in a rainbow color palette which intensity correlated with the estimated unfolding distance obtained from the SMFS FD-retraction curves.

**Theoretical HA-GPCR contour lengths**. Assuming that one amino acid is, on average, 0.4 nm long[35], we expect a single HA-β₂-AR unfolding at approximately 169 nm, HA-mGlu3-R at 355 nm, HA-2xβ₂-AR chimera at 315 nm, HA-µOR at 163 nm and HA-AT1a-R at 147 nm.

**Statistics and Reproducibility**. Statistical analysis was carried out using GraphPad Prism 9.1 software (GraphPad Software Inc.). Statistical tests and the number of independent experiments performed are indicated in the figure legends. A p value < 0.05 was considered as significant.

**Reporting summary**. Further information on research design is available in the Nature Research Reporting Summary linked to this article.

## Data availability

Data are available from the corresponding authors on reasonable request.

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

## Acknowledgements

We thank Anne-Marie Caminade and Jean-Pierre Majoral (LCC-CNRS UPR 8241, Toulouse) for providing us with the dendrimer used in this study for the AFM tip functionalization, and Marine Drapier together with Adrien Schaffner (INSA school, Toulouse) for their participation in the mathematical analysis of the AFM data. This work was supported by the Agence Nationale de la Recherche (ANR-17-CE11-0023-01 to C.G.) and the IDEX UNITI for the Transversality grant SMiRCH (UFTMIP: 2016-106-CIF-D-DRDV to E.D.).

## Author contributions

E.D. and C.G. conceived the study. E.D. designed and supervised the AFM-SMFS experiments. V.P. and C.G. analyzed all AFM-SMFS experiments. V.P. performed the GPCR functional experiments and constructed all figures. A.R. performed AFM-SMFS, ELISA experiments and Gaussian mixture analysis. S.A. performed the co-immunopre-cipitation/Western-blot experiments. V.L. performed some AFM-SMFS experiments. J.M.A. designed and supervised the statistical analysis of all AFM-SMFS data. S.V. wrote the R scripts for the Gaussian mixture analysis and the spatial representation of the receptor unfolding distances. E.T. supervised the chemical functionalization of the AFM tip. D.N. assisted in plasmid vector purification and cell culture and transfection. J.M.S. assisted in data processing and statistical analysis. C.G. designed experiments, analyzed data and wrote the manuscript with input from all the authors.

## Competing interests

The authors declare no competing interests.
