## [Peer Review File · Communications Biology]

Reviewers' comments:

Reviewer #1 (Remarks to the Author):

The manuscript describes, by means of single molecule force spectroscopy, the coexistence of different GPCR populations at the surface of mammalian cells, dependent on the GPCR oligomeric architecture, their expression and activity. The novelty of this work lies in (i) the characterization of GPCR unfolding events directly on living cells, rather than on purified systems as previously studied in the literature, and in (ii) the deepened investigation of the GPCR oligomeric populations at the cell surface which further the previous reports e.g. in TIRF and SIM studies. The manuscript is well written and provides a wealth of information. However, I have major concerns, essentially on the initial hypothesis underlying the whole work and conclusions that need to be properly addressed for the manuscript to be considered further. Please find my detailed comments below.

Major comments

1. The features shown in the curves provided in the extended data figure 1 do not resemble those of specific unfolding / adhesive events. The authors claim that they do not see the typical unfolding pattern of GPCR due to the complex interplay between the GPCR and the native membrane. Though I understand that purified systems and living ones would not rise similar features, at least should they observe a curvature in the adhesive event, typical of the stretching of the complex formed by the Ab and the tagged-GPCR. How do the authors assess the specificity of their adhesive events? Do they fit with e.g. the worm-like chain model? Their curves resemble more membrane tethers than specific unfolding of GPCR ... A control with, for instance, fixed cells would be appreciated to check if similar features are still observed or not, and to be confident of the following conclusions the authors rise based on those curves. Finally, given the lack of unfolding signature, the term should be replaced by unbinding for instance.

2. The authors should detail how the protomers of each specific GPCR receptor under investigation might be / are linked together. Indeed, it is not straightforward that a multimer would unbind at distance that is a multiple of the theoretical length of the monomer. If the molecules are not bound covalently to each other, how can it be that, after losing the contact, you are still able to unfold e.g. the second molecule in a dimer? Especially, the discussion on HA-mGlu3-R is really confusing in this sense, with a gaussian also found $\sim 77\text{nm}$ which is far below the GPCR length (355nm). I would expect, if multimeric states are indeed present at the cell surface, differences in the force distributions, rather than- or along with the rupture length distributions.

3. Assuming that the unbinding events reported in the manuscript are indeed specific to the GPCR receptor, and can be attributed to different oligomeric states, then differences should also be observed in terms of bond stiffness. It is indeed quite tempting to speculate that increasing the multimeric state of the receptor would change its stiffness. Though in a different context, C. Lo Giudice et al. (Nano Lett. 2020, 20, 5575-5582) have shown that upon ligand binding, conformational change of the G-protein-coupled glucagon receptor came along a stiffening of the binding complex. Such a stiffening might also occur before ligand-binding, in the case of multimers formation, for the receptor to express its more efficient state for further activity. The authors should report this information along with forces and rupture lengths to have a full picture of the mechanical signature of the receptors.

Specific comments along the text

- Section 1, p4 – criteria for FD curve inclusion-exclusion: based on their previous papers, authors only consider events with a rupture force $> 40\text{pN}$. First, it would be nice to see the force distributions associated with the rupture length distributions: is there any correlation between those two parameters? Second, the value of the rupture force cannot be an exclusive criterion for specificity (see my major comment 1).

- Section 2, p5– extended data figure 3: WTT-CHO cells transfected with an empty vector still show 18% of specific adhesion events, which is already quite high, though much lower than on

cells transfected with the appropriate positive vector. Can the author explain such discrepancies? They should add to this figure the distribution of the rupture length for the reader to compare this negative control to Fig.3. If no receptors are expressed at the cell surface, the force / rupture length signatures should be very different.

- Section 2, p5 – HA-mGlu3-R: “the unfolding distances were always slightly greater than the theoretical length”. A deviation of more than 100nm is not really small and can hardly originate from different interconnections.
- Section 2, figures 3-4: At least nine cells from three independent experiments were analyzed. The authors should at least provide one or two more examples of each condition in the extended data figure, especially for the spatial distributions on the map, for the reader to be fully confident on the nanoclusters formation with higher oligomeric states.
- Methods: the authors used a constant speed of 3µm/s. Have they tried to increase the speed and check for some influence on the GPCR signatures? Given the very soft nature of living mammalian cell surface, it would not be surprising to stretch the membrane and not the complex formed by the Ab and the GPCR at this very low speed

Reviewer #2 (Remarks to the Author):

The manuscript „AFM-single-molecule force spectroscopy unveils GPCR cell surface architecture” from Céline Galés’ lab presents a set of data from AFM-single-molecule force spectroscopy on living cells transiently transfected with HA-tagged GPCRs. By using an antibody-coated cantilever that binds to HA-tagged receptors they can literally pull them out of the cell membrane until rupture occurs. This allows precisely measuring the length of the pulled protein or protein complexes. For calibration of the length of the pulled protein they provide theoretical estimates for the length of a monomer, dimer or higher order oligomer. Following this line of reasoning, the authors observe some kind of spatial heterogeneity in GPCR surface expression including clusters of longer proteins and a rather homogenous distribution of monomers and dimers. This spatial organization is suggested to be receptor-specific, and dependent on expression levels and agonist stimulation.

Although the experiments appear to be well done, the results are clearly presented and the manuscript is very well written, I question the biological significance of this manuscript. There has been quite some debate in the field over the past 20 years of whether Class A GPCRs form dimers or not and whether these dimers (or higher order oligomers) have different functions than monomers. Whereas Class C GPCRs form obligate dimers, dimerization of Class A GPCRs appears to be much more transient. Based on recent literature that is also cited by the authors in their manuscript (Möller et al., Nat Chem Biol., 2020), dimerization occurs on the hundred millisecond timescale. This seems plausible considering the fact that GPCRs activation occurs within tens of milliseconds, G-protein activation in 500 ms, and cAMP accumulation on the second timescale. If GPCR oligomerization were to have a functional impact on GPCR signaling, it must occur on a similar timescale. In stark contrast, the authors of this manuscript have chosen a method with very low temporal resolution (in the minutes range) that does not appear powerful enough to report on the functional impact of physiologically-important transient dimer formation. Therefore, I consider the results of their experiments as biologically irrelevant.

In addition to this conceptual criticism, I have a number of more technical comments:

- 1) How do the authors make sure that the proposed GPCR dimers and higher-order oligomers are distinct from simply a higher receptor density in coated-pits (that are less mobile and probably easier to catch and pull). Moreover, this could explain that at – higher expression levels the b2AR – forms clusters that could present receptor crowding in clathrin-coated pits that is followed by increased internalization. It would be important to address this point experimentally.
- 2) Please provide proper experimental monomer and dimer controls to validate your method. Good monomeric controls could be the b1AR or the muOR. I am especially worried about the mGluR

data. It does not convince me that the applied method can reliably detect GPCR dimers. Please try other dimeric controls to increase power of the method.

3) There is no data on the actual nature of the proteins that are pulled out of the membrane. It would be crucial to show that their pulling approach specifically pulls receptor oligomers and not some kind of receptor complexes with accessory proteins. Would it be possible to do mass spec on the pulled proteins that stick to the cantilever?

Reviewer #3 (Remarks to the Author):

The manuscript of E. Dague et al. concerns a study in which the authors used the AFM operating in the force spectroscopy mode to map the distribution of GPCRs on living cells. The specificity of the interaction was obtained by functionalizing the AFM tip with anti-HA antibodies. The analysis of the force distance curves showing the unfolding of the receptors permitted to distinguish different populations (i.e. mono-, di, and tetramers) whereas the spots where unfolding events happened permitted to analyze the receptors distribution on the cellular surface. The technique is innovative and the topic highly interesting. The manuscript is well written and very comprehensive. It fully deserves according to me publication after minor modifications.

In the introduction :

- the term of pathophysiology should be replaced by pharmacology
- The authors should mention the time needed to record a single 16 x 16 pixels large image.
- The spot (i.e. above the nucleus, cytoplasm or periphery of the cells) on which the recordings were accomplished should be also specified

The upper right subfigure of fig 1a should be made larger. The chemical groups are barely visible.

DETAILED ANSWERS TO REVIEWER 1

Reviewer #1 (Remarks to the Author):

The manuscript describes, by means of single molecule force spectroscopy, the coexistence of different GPCR populations at the surface of mammalian cells, dependent on the GPCR oligomeric architecture, their expression and activity. The novelty of this work lies in (i) the characterization of GPCR unfolding events directly on living cells, rather than on purified systems as previously studied in the literature, and in (ii) the deepened investigation of the GPCR oligomeric populations at the cell surface which further the previous reports e.g. in TIRF and SIM studies. The manuscript is well written and provides a wealth of information. However, I have major concerns, essentially on the initial hypothesis underlying the whole work and conclusions that need to be properly addressed for the manuscript to be considered further. Please find my detailed comments below.

Major comments

→ 1. The features shown in the curves provided in the extended data figure 1 do not resemble those of specific unfolding / adhesive events. The authors claim that they do not see the typical unfolding pattern of GPCR due to the complex interplay between the GPCR and the native membrane. Though I understand that purified systems and living ones would not rise similar features, at least should they observe a curvature in the adhesive event, typical of the stretching of the complex formed by the Ab and the tagged-GPCR. How do the authors assess the specificity of their adhesive events? Do they fit with e.g. the worm-like chain model? Their curves resemble more membrane tethers than specific unfolding of GPCR ... A control with, for instance, fixed cells would be appreciated to check if similar features are still observed or not, and to be confident of the following conclusions the authors rise based on those curves. Finally, given the lack of unfolding signature, the term should be replaced by unbinding for instance.

ANSWER 1: The reviewer concern is justified. We don't perfectly understand how a specific unfolding / adhesive event in living cells should look like. However, the specificity of the anti-HA/HA adhesive events was extensively demonstrated both *in vitro* (characterization of HA/anti-HA specific binding using anti-HA-functionalized AFM tip and HA-functionalized surfaces) and *in vivo* (in HA-GPCR-expressing-WTT-CHO cells versus negative control/HA-GPCR non expressing cells) and detailed in our previous article published in 2015 (*Formosa C. et al. J Mol Recognit, 2015, DOI: 10.1002/jmr.2407*). Exactly the same specificity control (*living HA-GPCR-expressing-WTT-CHO cells versus HA-GPCR-non expressing cells*) was reproduced in the work presented in the current manuscript. The data presented in the present work are consistent with our previous work (*Formosa et al. J Mol Recognit, 2015*), demonstrating that such unfoldings are specific of cells expressing the HA-tagged beta 2-adrenergic receptor (HA- β_2 -AR). Actually, figure 3 of the same paper (*See below*) clearly demonstrates that the FD curve profiles are specific of the receptor-transfected cells (**i**), compared with **control WTT-CHO cells** (untransfected **g** or transfected with an "empty vector" **h**) but also do not present any similar adhesion forces (untransfected cells **d**, empty vector transfected cells **e**, receptor-transfected cells **f**). Similar untransfected control cells are classically used for AFM-SMFS experiments in living mammalian cells (*cf Knoop et al, Cell Chem Biol 2018 / DOI: 10.1016/j.chembiol.2018.02.006*).

In addition, to further strengthen that the FD profiles obtained with the anti-HA-functionalized-AFM-tip on WTT-CHO cells are specific of the unfolding of HA-tagged GPCRs expressed at the surface of the cells, we now provide in the **new Fig. 3c**, and **Extended Data Fig. 8** representative maps of the rupture force obtained in WTT-CHO cells transfected with different HA-GPCRs (HA- β_2 -AR, HA-mGlu3-R, HA-2x β_2 -AR chimera) compared with those of **control** WTT-CHO cells, which are transfected with an empty vector (*to take into account the influence of the transfection agent on the cell surface*) and therefore do not express the HA-GPCR at their surface. As you can see in **these new figures**, this comparison definitively demonstrates that the force profile is specific of the HA-tagged GPCR with higher forces depicted in cells expressing HA-GPCRs compared with control cells.

Regarding the observation of “**a curvature in the adhesive event, typical of the stretching of the complex formed by the Ab and the tagged-GPCR**”. The final stretching described by the reviewer is usually attributed to the elongation of the linker in between the AFM tip and the antibody. In the present work, the functionalization strategy is based on the use of a dendrimer that serves as linker. This round shaped molecule (the 4th generation dendrimer) is not stretchable as a linear linker can be. This specificity of the dendrimer has been previously described in a patent (*Dague E., Trevisiol E., and Jauvert E. 2012-Pointe de microscope à force atomique modifiée et biomodifiée, N° de publication FR2965624 A1; Date de publication 2012-04-06*) and an article (*Jauvert E., Dague E., Severac M., Ressler L., Caminade A-M., Majoral J-P., and Trévisiol E. 2012 Probing single molecule interactions using bio-functionalized dendritips; Sensors and Actuators, 168, 436-441*).

Regarding the fitting with e.g. the worm-like chain model. The reviewer comment would be interesting in the context of purified proteins. To date, biophysical models like the WLC model are

useful to calculate the contour length of folded proteins. It works perfectly well in the context of purified $\beta 2$ -AR reconstituted into liposomes for example (Zocher *et al.* PNAS 2012). However, in our case, we do not expect the WLC model to fit our force profile at all. We are working on living cells and we do not expect to unfold a “pure” HA-tagged protein. The initial HA-tagged receptor pulled by the anti-HA antibody most probably comes with lipids from the membrane, but also other transmembrane proteins well known to interact with GPCRs and regulate their activity (i.e. Receptor-Activity Modifying Proteins-RAMP proteins, etc...). Thus, as shown below (fitting in green), the WLC model completely fails to fit with the recorded data. None of the HA-GPCR examined in this study were fitting with this WLC model. The use of the WLC fitting has been cited in the results section in the revised manuscript.

Regarding the fact that our curves could resemble more membrane tethers than specific unfolding of GPCR as stated by the reviewer. Membrane tethers are readily identifiable on our FD curves and are now presented in the **new Extended data Fig. 2**. Moreover, as shown, tethers are never pulled out alone but always following a specific HA/anti-HA adhesive event/unfolding distance, most probably indicative of the unfolding of HA-GPCR together with membrane lipids (which is not surprising based on the high number of crystallography studies showing the direct interactions between plasma membrane lipids and the amino acids of GPCRs).

Moreover, if unfolding distances reflected membrane tethers and were not specific GPCRs unfolding as speculated by the reviewer, then the number of adhesive events would be identical between control cells/transfected with the empty vector and cells transfected with the HA-GPCR, which is not the case since much higher number of adhesive events are recorded in HA-GPCR-expressing cells compared to the control cells.

The reviewer proposed a control with, for instance, fixed cells to check if similar features are still observed or not and to be confident of the following conclusions we rise based on those curves. As mentioned above, our control of specificity was to compare FD curves obtained in living WTT-CHO cells-transfected with a HA-GPCR-encoding vector (*HA-GPCR-expressing WTT-CHO cells*) versus those obtained in living WTT-CHO cells-transfected with an “empty” encoding vector (*WTT-CHO cells with no HA-GPCR at the cell surface = CONTROL cells*). Indeed, the use of fixatives (such as formaldehydes) would have introduced a technical artifact in our study examining GPCR organization at the surface of living cells (i.e. GPCR oligomerization) since they are well known to

promote protein cross-linking together with delipidation of the membrane. As cross-linking fixatives form chemical bonds between molecules, they are definitively not indicated to assess the native oligomeric organization of GPCR at the cell surface. Hence, living versus fixed cells could not have similar biophysical properties. In cell biology, the technical challenge relies precisely on stopping the use of fixatives to obtain a more real picture of the cell. Thus, the originality of our AFM study relies really here on the use of living cells.

The reviewer proposed, **given the lack of unfolding signature, to replace the term by unbinding for instance**. We do not agree since “Unfolding” events in AFM-SMFS are representative of adhesive events while the term “unbinding”, as proposed by the reviewer, could be interpreted as the non-adhesive events and not as the unbinding between the anti-HA-functionalized tip and the HA-tagged GPCR. Indeed, all these adhesive and non-adhesive events are indicated in the pie charts of the figures. Moreover, these adhesive/non-adhesive events obtained in WTT-CHO cells expressing or not the different GPCRs are quite informative regarding the specificity of the adhesive events. Indeed, if adhesive events were non-specific, the number of adhesive events would be similar in all experimental conditions whatever the GPCR transfected or not, which is not the case since β_2 -AR displays much lower number of adhesive events on a $3 \times 3 \mu\text{m}^2$ map as well illustrated on the spatial maps compared to the other GPCRs tested. Further comforting the specificity of the unfolding curves obtained in all HA-GPCR transfected cells, the control with no expression of HA-GPCR at the cell surface displays the lower number of adhesive events (18%) among all our experimental conditions. These 18% adhesive events are not surprising and most likely represent the non-specific adhesion of the HA-antibody at the cell surface that is not specific of the AFM technique but commonly observed in all biological experiments using antibodies in living cells or tissues and that usually introduced a “blocking step” with BSA or other reagents to block the non-specific sites on the cell surfaces and decrease the background non-specific signal (that is not possible in AFM since BSA would stick the tip). Finally, if adhesive events were all non-specific, unfolding lengths would be also similar between different transfected WTT-CHO cells which is not the case since the control cells (no expression of HA-GPCR at the cell surface) never display long unfolding distances compared to the GPCR conditions.

→ 2. The authors should detail how the protomers of each specific GPCR receptor under investigation might be / are linked together. Indeed, it is not straightforward that a multimer would unbind at distance that is a multiple of the theoretical length of the monomer. If the molecules are not bound covalently to each other, how can it be that, after losing the contact, you are still able to unfold e.g. the second molecule in a dimer? Especially, the discussion on HA-mGlu3-R is really confusing in this sense, with a gaussian also found ~77nm which is far below the GPCR length (355nm). I would expect, if multimeric states are indeed present at the cell surface, differences in the force distributions, rather than- or along with the rupture length distributions.

ANSWER 2: Protomers of GPCRs are linked together through hydrophobic interactions and this is a common feature of most transmembrane proteins. These hydrophobic interactions confer very high stability to the GPCR oligomers which impart for their resistance to SDS denaturation (*See review Bouvier M, Nat Rev Neurosci. 2001 Apr;2(4):274-86. doi: 10.1038/35067575*) and visualization of GPCR oligomers in western-blot analysis. This is how the concept of GPCR oligomerization was originally discovered and further confirmed using different techniques. For a better understanding by broad readers, we have now reintroduced this notion in the revised version of our manuscript (**Page 6-7**).

Moreover, as explained in the discussion part of the original manuscript (**Page 13**, *see reference 23, Cordomi A, et al, Trends Biochem Sci. 2015 Oct;40(10):548-551. doi: 10.1016/j.tibs.2015.07.007*), the

interface of dimerization/oligomerization is not unique and highly differs in between GPCRs (but also in between activation states of a same receptor and between cell types). It follows that depending on the GPCR and the transmembrane domains involved in the interactions, we could have a more linear interaction between two protomers as proposed for the β_2 -AR in the current study for which unfolding distances are close to the multiple of the length of one protomer while a more intertwined connection between transmembrane domains of two protomers was proposed for the mGlu3-R, thus leading to the unfolding of packed oligomers.

Regarding the HA-mGlu3-R with a Gaussian also found ~ 77 nm which is far below the GPCR length (355 nm). We totally agree with the reviewer but this 77 nm Gaussian represent less than 5 % of the total adhesive curves and was thus not taken into account for the biological interpretation of the data as stated in the first version of our manuscript.

Now, in the revised version, we have introduced a new control for GPCR dimer consisting in a fusion protein that connects two protomers of β_2 -AR (HA- β_2 -AR- β_2 -AR also called HA-2x β_2 -AR chimera) with a theoretical length of 315 nm that is closed to the length of one mGlu3-R protomer (355 nm). It is interesting to note that in WTT-CHO cells expressing the HA-2x β_2 -AR chimera, we also found a Gaussian at 63 nm (See New Fig. 4c) close to that of found at 77 nm with the mGlu3-R and similarly accounting for less than 5% of all the adhesive events. Such a Gaussian was never observed with other GPCRs with lower theoretical contour lengths. Whether this peculiar feature of long GPCRs reflects abortive rupture due to the insertion of these long transmembrane proteins in specific nanodomains of the plasma membrane difficult to extract is unknown but could be envisioned.

Regarding the expectation of the reviewer to have multimeric states of the GPCRs correlated with differences in the force distributions, rather than- or along with the rupture length distributions. Spatial maps indicative of the force rupture obtained by SMFS on the $3 \times 3 \mu\text{m}^2$ area have been now established. Correlations were also calculated between the unfolding distance and the corresponding rupture force for the HA- β_2 -AR, the HA-mGlu3-R as well as for the dimeric control HA-2x β_2 -AR chimera. However, as you can see, we didn't find any correlation between unfolding distances and rupture forces. In the revised manuscript, we have introduced representative adhesion maps as a new Fig. 3c, d and Extended Data Fig. 8 while the correlation was cited in the results part (*End last paragraph, page 7*).

→ 3. Assuming that the unbinding events reported in the manuscript are indeed specific to the GPCR receptor, and can be attributed to different oligomeric states, then differences should also be observed in terms of bond stiffness. It is indeed quite tempting to speculate that increasing

the multimeric state of the receptor would change its stiffness. Though in a different context, C. Lo Giudice et al. (Nano Lett. 2020, 20, 5575-5582) have shown that upon ligand binding, conformational change of the G-protein-coupled glucagon receptor came along a stiffening of the binding complex. Such a stiffening might also occur before ligand-binding, in the case of multimers formation, for the receptor to express its more efficient state for further activity. The authors should report this information along with forces and rupture lengths to have a full picture of the mechanical signature of the receptors.

ANSWER 3: The reviewer is absolutely right and the idea is excellent. This analysis would give rise to a very nice biophysical work. Nevertheless, this is a completely different story and the implementation of this very pertinent analysis would bring confusion to the paper demonstration. To be clear, the analysis suggested by the reviewer is of high interest but beyond the scope of the present work. The paper scope is more biological, i.e. understanding the architectural organization of GPCRs in native living mammalian cells.

However, to reinforce the demonstration that the unfolding events are specific to a specific oligomeric state of the GPCR, we have now introduced in the revised version of our manuscript a dimeric control consisting in a fusion protein between two β 2-AR protomers (HA-2x β 2AR chimera) (*see answer 2-penultimate paragraph*). Results obtained with the β 2-AR fusion protein (dimeric control) (**presented in the new Fig. 4c, g, k**) indicate main rupture lengths with Gaussian peak perfectly fitting with the theoretical contour length of a protomer (~350 nm), a dimer (~610 nm), a trimer (~1004 nm) and a pentamer (~1611 nm), thus clearly demonstrating that the present AFM-SMFS method reliably detects GPCR oligomers

Regarding the paper of C. Lo Giudice et al. (Nano Lett. 2020, 20, 5575-5582) highlighted by the reviewer. In the referenced article, the authors are demonstrating that the interaction of the ligand with its receptor induces a conformational and mechanical modification. It is technically very impressive but in principle, it is the way it works. When a ligand binds to its receptor, this promotes a change in the receptor conformation and an ensuing intracellular signaling cascade.

In our case, this basic principle is not operational as we are not stimulating the receptors with a ligand. It is obvious that the conformation of a dimer is different of the conformation of a monomer and therefore also very likely that the mechanical properties of a monomer are different from a dimer. Again, the scope of the paper is initially not related to the biophysical properties of the oligomers but more to the understanding of the architectural organization of GPCRs at the surface of living cells. Therefore, this very interesting comment but analysis is beyond the scope of the present work

Finally, **to reinforce that the unbinding events reported in the manuscript are indeed specific to the GPCR receptor, and can be attributed to different oligomeric states**, we proceeded to the analysis of GPCR oligomerization through another technical approach, *i.e.* the biochemical Immunoprecipitation (IP)/Western-blot. The IP experiments were performed using an anti-HA antibody for the 2 GPCRs HA- β 2-AR and HA-mGlu3-R (*Also for the HA- μ OR a new GPCR in the revised manuscript*) and at basal state in similar conditions to that were used for the AFM study to compare the results with the AFM-unfolding distances. These new data are presented in **the new Extended Data Fig. 7** of the manuscript. As shown below, we identified mainly monomers and high-order oligomer receptor populations for the HA- β 2AR but also HA-mGlu3R, thus correlating perfectly with the main Gaussian peaks obtained with the unfolding distances obtained with these receptors.

Specific comments along the text

• Section 1, p4 – criteria for FD curve inclusion-exclusion: based on their previous papers, authors only consider events with a rupture force > 40pN. First, it would be nice to see the force distributions associated with the rupture length distributions: is there any correlation between those two parameters? Second, the value of the rupture force cannot be an exclusive criterion for specificity (see my major comment 1).

ANSWER 4: These adhesion forces (force maps) and force / length correlations have been done but no correlation was observed (See answers 2 and 3). Also, in the revised manuscript, we have introduced the adhesion maps as a new Fig. 3c, d and Extended Data Fig. 8 while the correlation was only cited in the results part (End last paragraph, page 7).

Regarding the value of the rupture force that cannot be an exclusive criterion for specificity as stated by the reviewer. The reviewer is pointing out a fundamental drawback of AFM analysis. It is never possible to have a direct proof, evidence, that a force profile is related to a specific event. We have thus to rely on indirect evidences and negative/positive controls.

All the work presented in the current manuscript relies on our previous publication, Formosa *et al.* 2015, demonstrating that the unfoldings are specific to cells expressing the HA-tagged GPCR, and as stated

in this previous manuscript and in **answer 1 and 5**, we have now also established a number of new negative / positive controls that strengthen the specificity of the unfolding distances in HA-GPCR-expressing WTT-CHO cells in the revised version of the manuscript. We have also better detailed in the “Methods” section our criteria for FD curve inclusion-exclusion (**Inclusion:** Adhesion forces $\geq 40\text{pN}$; **Exclusion:** disturbances in the approach curve, tethers, unidentifiable baseline).

• **Section 2, p5– extended data figure 3: WTT-CHO cells transfected with an empty vector still show 18% of specific adhesion events, which is already quite high, though much lower than on cells transfected with the appropriate positive vector. Can the author explain such discrepancies? They should add to this figure the distribution of the rupture length for the reader to compare this negative control to Fig.3. If no receptors are expressed at the cell surface, the force / rupture length signatures should be very different.**

ANSWER 5: (*refer also to answer 1 last paragraph*) These 18 % adhesive events are not surprising and most likely represent the non-specific adhesion of the HA-antibody at the cell surface that is not specific of the AFM technique but commonly observed in all biological experiments using antibodies (immunofluorescence, western-blot...) in cell culture or tissues and that usually introduced a “blocking step” with BSA or other reagents to block the non-specific sites on the cell surfaces and decrease the background non-specific signal (that is not possible in AFM since BSA would stick the tip). Importantly and as underlined by the reviewer, this 18% of adhesive events in the negative control **is much lower than other conditions expressing the HA-GPCRs**. Indeed, quantifications of adhesive/non-adhesive events in HA-tagged GPCR-expressing WTT-CHO cells indicated **from 44 %** (*the lowest for HA- β_2 -AR^{high} that we demonstrated to be related to its ability to continuously internalize in the absence of ligand*) **to more than 90 % of adhesive events** for all the other GPCRs tested at resting state in the manuscript (HA-mGlu3-R, HA-AT1a-R, **thus demonstrating the high specificity of the functionalized AFM tip for the HA-GPCRs expressed at the cell surface.**

Because of the very low number of rupture lengths in the negative control, we couldn't proceed to a statistical analysis through Gaussian distribution. However, we have now introduced in the revised manuscript (**New Fig. 3c,d and new Extended Data figure 8**) at least 3 representative spatial maps of the rupture forces obtained in control cells (from different cells/experiments) compared to those obtained in HA-GPCR-expressing cells. Again, as shown, and in agreement with our previous work (*Formosa et al, J Mol Recognit. 2015*), we show a marked difference between force maps of control cells and HA-GPCR-expressing cells (higher forces).

• **Section 2, p5 – HA-mGlu3-R: “the unfolding distances were always slightly greater than the theoretical length”. A deviation of more than 100nm is not really small and can hardly originate from different interconnections.**

Answer 6: We completely agree with the reviewer. This was already discussed in our previous version of our manuscript in the discussion section (**page 13**). Indeed, the high standard deviation of the Gaussian (rupture lengths) observed in the HA-mGlu3-R compared with HA- β_2 -AR could readily affect the accuracy of the Gaussian peak. Taken together with the additional hypothesis of different interconnections between promoters depending on the GPCR (see answer 2), this could account for the 100 nm deviation.

• **Section 2, figures 3-4: At least nine cells from three independent experiments were analyzed. The authors should at least provide one or two more examples of each condition in the extended data figure, especially for the spatial distributions on the map, for the reader to be fully confident on the nanoclusters formation with higher oligomeric states.**

Answer 7: As suggested by the reviewer, two more examples of the spatial distribution of the rupture lengths have been introduced in the **new Extended Data Figure 4**. As you can see, different spatial maps for one condition (control or HA-GPCR) are highly similar in between cells and experiments, highlighting the reproducibility of the adhesive/non-adhesive events as well as the distances obtained in each condition.

• Methods: the authors used a constant speed of 3µm/s. Have they tried to increase the speed and check for some influence on the GPCR signatures? Given the very soft nature of living mammalian cell surface, it would not be surprising to stretch the membrane and not the complex formed by the Ab and the GPCR at this very low speed

ANSWER 8: No, we did not check the influence of the tip speed on the GPCR signature. We monitored the influence of the Loading Rate on the HA-anti-HA recognition *in vitro* in our previous paper (*Formosa et al, J Mol Recognit. 2015*). On the contrary, in this current manuscript, we tried to stretch the GPCR at constant force, with no success in terms of a specific GPCR signature. As mentioned in answer 2 (paragraph 5), tethers are easily identifiable on the FD curves. Moreover, if the adhesive events reflected membrane tethers and not the HA-anti-HA complex, the number of adhesive events would be identical between control cells/transfected with the empty vector and cells transfected with the HA-GPCR, which is not the case since a much higher number of adhesive events were recorded in HA-GPCR-expressing cells compared to the control cells.

DETAILED ANSWERS TO REVIEWER 2

Reviewer #2 (Remarks to the Author):

The manuscript „AFM-single-molecule force spectroscopy unveils GPCR cell surface architecture” from Céline Galés’ lab presents a set of data from AFM-single-molecule force spectroscopy on living cells transiently transfected with HA-tagged GPCRs. By using an antibody-coated cantilever that binds to HA-tagged receptors they can literally pull them out of the cell membrane until rupture occurs. This allows precisely measuring the length of the pulled protein or protein complexes. For calibration of the length of the pulled protein they provide theoretical estimates for the length of a monomer, dimer or higher order oligomer. Following this line of reasoning, the authors observe some kind of spatial heterogeneity in GPCR surface expression including clusters of longer proteins and a rather homogenous distribution of monomers and dimers. This spatial organization is suggested to be receptor-specific, and dependent on expression levels and agonist stimulation.

Although the experiments appear to be well done, the results are clearly presented and the manuscript is very well written, I question the biological significance of this manuscript. There has been quite some debate in the field over the past 20 years of whether Class A GPCRs form dimers or not and whether these dimers (or higher order oligomers) have different functions than monomers. Whereas Class C GPCRs form obligate dimers, dimerization of Class A GPCRs appears to be much more transient. Based on recent literature that is also cited by the authors in their manuscript (Möller et al., Nat Chem Biol., 2020), dimerization occurs on the hundred millisecond timescale. This seems plausible considering the fact that GPCRs activation occurs within tens of milliseconds, G-protein activation in 500 ms, and cAMP accumulation on the second timescale. If GPCR oligomerization were to have a functional impact on GPCR signaling, it must occur on a similar timescale. In stark contrast, the authors of this manuscript have chosen a method with very low temporal resolution (in the minutes range) that does not appear powerful enough to report on the functional impact of physiologically-important transient dimer formation. Therefore, I consider the results of their experiments as biologically irrelevant.

ANSWER 1: We think that the reviewer likely misinterpreted the time scale of the unfolding measurements. We have now better detailed this aspect in the beginning of the “results” section of our revised manuscript (*page 4*) and in the **new Extended Data Fig. 1**.

As indicated in the first paragraph of our manuscript, we indeed scanned a $3 \times 3 \mu\text{m}^2$ area at the top of the cell thus allowing the recording of a raster of 16×16 (256) FD curves) **taking about 10 minutes for the whole raster** (*briefly mentioned in the methods section of our original manuscript*). **However, one should not confuse the time needed to obtain the whole map with the time needed for one measurement, i.e. for the acquisition of a single force curve that is indeed less than 500 ms.**

On other words, for every pixel among the total 256 pixels, one approach-retraction cycle of the cantilever is recorded. As shown below on a Force / Time curve, the reviewer can better appreciate the timescale of one FD curve recording , with **1/** the approach of the tip to the cell surface, **2/** the **200 ms** pause of the tip at the cell surface to optimize the probability of a single interaction between the HA antibody and the HA-GPCR, **3/** the retraction of the tip from the surface taking **less than 500 ms** (thus agreeing with the time scale of activation) with the unfolding events in case of the existence of adhesive forces between the tip and the surface and **4/** the rupture of the tip / surface interaction and return to the baseline.

Compared to the TIRF microscopies studies mentioned by the reviewer which is dedicated to the study of the dynamics/mobility of GPCR oligomers at the cell surface on living cells, this SMFS-AFM technique examined the equilibrium of the different GPCR populations at the cell surface within the 500 ms range and not the dynamics in real-time.

To date, the existence of different populations of a GPCR at the cell surface mainly relies on the same technique, *i.e.* the TIRF microscopy which is not without drawbacks like every technique, with for instance obligation to have very low expression of the fluorescent proteins, study only of the cell surface in contact with the glass with thus very high stiffness which can highly influence the organization/mobility of GPCRs at the cell surface, and finally the assumption that the control transmembrane protein, usually CD28 or CD86, is solely in a monomeric form. This is thus important in the field, to ascertain such observations by multiplexing the techniques.

In addition to this conceptual criticism, I have a number of more technical comments:

1) How do the authors make sure that the proposed GPCR dimers and higher-order oligomers are distinct from simply a higher receptor density in coated-pits (that are less mobile and probably easier to catch and pull).

ANSWER 2: We think there is a misunderstanding of the reviewer since higher receptor density in microdomains such as coated pits does not mean receptor oligomerization. If receptors were clustered into coated pits, as stated by the reviewer, it would induce a proximity but not a physical interaction between promoters of a GPCR. Indeed, specific domain interactions between promoters are needed to interact and trigger receptor oligomerization. Hence, higher receptor density reflecting higher number of receptor monomers should be reflected in AFM as higher number of monomer-corresponding distances but not as oligomer-corresponding distances.

In addition, it is now well established for many years that receptor clustering in clathrin-coated pits cannot be associated with receptor crowding. To the contrary, agonist binding to the GPCR triggers receptor activation and β -arrestin recruitment, thereby promoting receptor clustering in clathrin-coated pits (through β -arrestin binding to clathrin-AP2) and further internalization. Thus, clathrin-dependent receptor endocytosis is a fine-tuned regulated process reflecting an activated system. By opposition, in our SMFS-experiments, high-order oligomers are observed for all our GPCRs tested at steady state in the absence of ligand. It is thus unlikely that all the GPCR tested that did not display constitutive activity (except for the β_2 -AR with constitutive activity), are confined in coated pits at resting state.

Now, to better ascertain that main unfolding distances (Gaussian peaks) in AFM-SMFS truly reflect different GPCR oligomeric populations, in parallel to the AFM experiments, we performed

immunoprecipitation experiments (under exactly similar transfection conditions in WTT-CHO cells) followed by western-blot to thus examine the existence of the different GPCR oligomeric states through another technique. As the reviewer can see, all the tested GPCR exist as different oligomeric states at steady state in these WTT-CHO cells at similar expression level to that in the SMFS experiments, thus reinforcing the specificity of the AFM-SMFS technique to unfold GPCR oligomeric populations. These data have been now introduced in the new **Extended Data Fig. 6d et 7**.

Moreover, this could explain that at – higher expression levels the b2AR – forms clusters that could present receptor crowding in clathrin-coated pits that is followed by increased internalization. It would be important to address this point experimentally.

ANSWER 3: (see also answer 2)) It is very unlikely that higher expression levels of β_2 -AR induced the formation of receptor crowding in clathrin-coated pits since it should also be the case for all the GPCRs tested. However, only β_2 -AR **displays** a high level of non-adhesive events that we demonstrated in the study to rely on the constitutive activity of this receptor on the internalization pathway (*See sucrose experiment to block the internalization process*). This result is completely in agreement with the well-known high constitutive activity of the β_2 -AR that we also demonstrated in our experimental conditions by monitoring basal cAMP production in HA- β_2 -AR-expressing WTT-CHO cells (Figure 2).

2) Please provide proper experimental monomer and dimer controls to validate your method. Good monomeric controls could be the b1AR or the muOR. I am especially worried about the mGluR data. It does not convince me that the applied method can reliably detect GPCR dimers. Please try other dimeric controls to increase power of the method.

ANSWER 4: As suggested by the reviewer, we have now tested **HA- μ OR** as a **monomeric control** but also a **dimeric control** consisting in a fusion protein that connects two protomers of β_2 -AR (**HA-2x β_2 -AR chimera**) with a theoretical length of 315 nm that is close to the length of one mGlu3-R protomer (355 nm).

Interestingly, **results obtained with the HA-2x β_2 -AR chimera** (dimeric control) (presented in the **new Fig. 4c, g, k**) indicate main rupture lengths with Gaussian peaks perfectly fitting with the theoretical length of a chimera monomer (350 nm), a dimer (610 nm), a trimer (1004 nm) and a pentamer (1611 nm). This result highly reinforces the notion that the SMFS method reliably detects GPCR oligomers

It is also interesting to note that in WTT-CHO cells expressing the HA-2x β_2 -AR chimera, we also found a Gaussian at 63 nm close to that of found at 77 nm with the mGlu3-R and also represents less than 5 % of all the adhesive events. Such a Gaussian was never observed with other GPCRs with lower theoretical lengths. Whether this peculiar feature of long GPCRs reflects abortive rupture due to the insertion of these long transmembrane proteins in specific nanodomains of the plasma membrane difficult to extract is unknown but could be envisioned.

While it is easy to design a dimeric control by engineering a fusion protein with longer contour length (i.e. HA-2x β_2 -AR chimera), it is really more challenging to find a proper monomeric control. Indeed, receptor oligomeric state was highly dependent on receptor expression and cell types. Hence, while β_1 -AR is described to be prevalently monomeric at low densities, it also tends to forms an increasing number of dimers at higher densities (*Calebiro et al. PNAS. 2013*). We chose the μ -opioid receptor (μ OR) as a monomeric control as it was mostly reported as monomer, at least at low expression level and in CHO cells (*Möller et al. Nat Chem Biol. 2020*). **Results with the HA- μ OR are presented in the new Extended Data Fig. 6**. As you can see (**new Extended Data Fig. 6a**), in the WTT-CHO

cells, while we detect a Gaussian peak (188 nm) close to the theoretical length of the monomer (163 nm), the main Gaussian peaks were detected for a population close a HA- μ OR dimer (398 nm) and high-order oligomers (751 nm and 1249 nm). This suggest that in our experimental conditions in WTT-CHO cells (that are different from the CHO cells classically used in TIRF microscopy), HA- μ OR exhibits different oligomeric states at the cell surface.

This result was further confirmed by biochemical Immunoprecipitation/Western blot analysis of the HA- μ OR in WTT-CHO cells (**new Extended Data Fig. 6d**) in which we clearly found that HA- μ OR exists as high-order oligomeric states together with the monomeric form. Noteworthy, the high oligomeric forms of this receptor predominate over the monomer even at lowest expression level of the HA- μ OR, thus completely corroborating the Gaussian analysis of the unfolding lengths obtained by AFM-SMFS depicting a minority of monomer (188 nm) while dimer- and high-order oligomer-corresponding lengths predominate.

It is important to note that the recent TIRF microscopy study describing HA- μ OR as a main monomer was tested only at very low expression level and in a specific cell line (CHO cells). Importantly, under these conditions, ~ 10% of HA- μ OR were detected as a dimer (*Möller et al. Nat Chem Biol. 2020*). More recent studies using TIRF or FRET experiments (*Asher W.B. et al, Nat Methods 2121, 397-405 / Cevocha K, Cell Mol Life Sciences, 2021*) have similarly examined the μ OR oligomerization state but in the same cell line, i.e. the CHO cells and again at low expression level (for technical considerations), and concluded on similar conclusion on the existence of μ OR essentially as a monomeric form at the cell surface. However, the oligomeric signature of a receptor is known to be dependent on its expression level (*Calebiro et al. PNAS. 2013*), its activation state (*Möller et al. Nat Chem Biol. 2020*) but also on the cell environment (lipids in the membrane, anchorage to cortical action...). Such impact of lipid composition on μ OR oligomerization has been recently highlighted in modeling studies (*Marino KA, PLoS Comput Biol. 2016 Dec 13;12(12):e1005240; DOI: 10.1371/journal.pcbi.1005240*) **It thus impossible to assign an unique oligomeric state signature to a receptor since this will be highly cell-dependent.** In our current study, we used a different cell line (WTT-CHO cells) that we described in the Methods section and that is different from CHO cells.

During the revision process, to confirm that the AFM-SFMS can truly be used to unfold and depict GPCR oligomerization, we have also conducted similar IP experiments to that performed for the μ OR but for 2 other GPCRs used in the beginning of the manuscript, i.e. the HA- β_2 -AR and the HA-mGlu3-R and in similar conditions to that were used for the AFM study to compare the results with the AFM-unfolding distances. These new data are presented in **the new Extended Data Fig. 7** of the manuscript. As shown below, we identified multiple HA- β_2 -AR monomers (different receptor glycosylation, phosphorylation...) and several high-order oligomers but to a lower extent (**Extended Data Fig. 7a**), correlating with the main unfolding distances obtained with this receptor (**Fig. 4a**), while high proportions of both monomers and high-order populations were depicted for the HA-mGlu3-R (**Extended Data Fig. 7b**) in agreement with the main Gaussian peaks obtained with the unfolding distances associated with this receptor (**Fig. 4b**).

3) There is no data on the actual nature of the proteins that are pulled out of the membrane. It would be crucial to show that their pulling approach specifically pulls receptor oligomers and not some kind of receptor complexes with accessory proteins. Would it be possible to do mass spec on the pulled proteins that stick to the cantilever?

ANSWER 5: This would be wonderful but unfortunately SMFS-AFM experiments typically measures **the rupture** between the anti-HA antibody fixed on the AFM-tip and the HA-tagged GPCR (see Fig. 1b, right panel), meaning that the protein unfolded are never completely pulled out of the plasma membrane and thus don't stick to the AFM tip.

DETAILED ANSWERS TO REVIEWER 3

Reviewer #3 (Remarks to the Author):

The manuscript of E. Dague et al. concerns a study in which the authors used the AFM operating in the force spectroscopy mode to map the distribution of GPCRs on living cells. The specificity of the interaction was obtained by functionalizing the AFM tip with anti-HA antibodies. The analysis of the force distance curves showing the unfolding of the receptors permitted to distinguish different populations (i.e. mono-, di, and tetramers) whereas the spots where unfolding events happened permitted to analyze the receptors distribution on the cellular surface. The technique is innovative and the topic highly interesting. The manuscript is well written and very comprehensive. It fully deserves according to me publication after minor modifications.

In the introduction :

- the term of pathophysiology should be replaced by pharmacology

ANSWER 1: the term “Physiopathology” was introduced here to specify that GPCRs constitute the main drug targets for the treatment of a large array of diseases. “pharmacology” would not be consistent with this notion.

- The authors should mention the time needed to record a single 16 x 16 pixels large image.

ANSWER 2: All parameters related to the kinetics of the FD curves recording have been now detailed at the beginning of the “results” section (**page 4**) and in the **new Extended Data Fig. 1**. As you will see, the time needed to record a single 16 x 16 pixels large image is ~10 min.

- The spot (i.e. above the nucleus, cytoplasm or periphery of the cells) on which the recordings were accomplished should be also specified

Answer 3: More precisely, we probed the cell surface stretched between the nucleus and the Petri dish. This was already illustrated in the original version of our manuscript in the **Fig. 1c** but was now also specified in the “results” section on **page 4**.

The upper right subfigure of fig 1a should be made larger. The chemical groups are barely visible.

Answer 4: This has been done as shown in the **new figure 1a** in the revised manuscript.

Reviewers' comments:

Reviewer #1 (Remarks to the Author):

In their revised manuscript, Dague et al. provided a number of additional experiments that further illustrate their message. Especially, the co-immunoprecipitation / Western blot experiments clearly confirm the presence of high GPCR oligomeric states at the cell surface, in agreement with the unfolding distances reported in AFM measurements. Additionally, the control with the HA-2xb2-AR chimera reinforces the results obtained on the initial GPCR.

I thank the authors for their very detailed and precise answers to my initial comments, the additional data they have included in the present manuscript and the rebuttal letter to strengthen their conclusions and the time consuming (I can imagine!) extra experiments / controls they have performed to convince me and the reader of the final message, i.e. the presence of receptor- and activity dependent distinct GPCR oligomeric state at the cell surface, dictating their spatial organization and, thus, potential ligand binding activity. I thus consider this manuscript as appropriate for publication with strong statistical & experimental work furthering our understanding of GPCR complexity.

I just have a very minor comment concerning typos in the introduction section: just after reference 5, you have two repetitions in the sentence ("will greatly depend" and "both the lipid composition of the plasma membrane").

Marion Mathelié

Reviewer #2 (Remarks to the Author):

The authors have made a lot of effort to address all of my comments to the original version of their manuscript. I feel that the manuscript has improved a lot both in terms of additional experiments that support the original conclusions of the manuscript as well as the clarity in writing.

Nevertheless, there are some remaining points that are puzzling and require further attention:

1) The manuscript still lacks a proper monomeric control. This can be either a GPCR, of course, but any other membrane protein that is strictly monomeric would be fine. This control is essential as it would prove that the method used by the authors is able to exclusively detect monomers as well. Moreover, this control experiment would rule out the possibility that this method is 'oligomeric-prone', i.e. that it tends to overestimate the amount of dimers or higher-order oligomers.

2) The new results on the μ -OR are interesting, however, it should be discussed more extensively why the authors' results are so different from what other labs have observed. Along this line it should be considered that μ -ORs have been reported to be essentially monomeric not only using single-molecule densities, but also by molecular brightness techniques (and FRET acceptor photobleaching) that are typically performed at much higher expression levels. As WTT-CHO cells and CHO-K1 cells are similar, this cannot be the only reason for the highly divergent results.

3) The mGluR3 is a Class C GPCR and an obligate dimer (or higher-order oligomer). How do the authors explain and justify that they observe a large population of monomeric receptors (39%, Figure 4b) in their measurements?

DETAILED ANSWERS TO REVIEWER 1

Reviewer #1 (Remarks to the Author):

In their revised manuscript, Dague et al. provided a number of additional experiments that further illustrate their message. Especially, the co-immunoprecipitation / Western blot experiments clearly confirm the presence of high GPCR oligomeric states at the cell surface, in agreement with the unfolding distances reported in AFM measurements. Additionally, the control with the HA-2xb2-AR chimera reinforces the results obtained on the initial GPCR.

I thank the authors for their very detailed and precise answers to my initial comments, the additional data they have included in the present manuscript and the rebuttal letter to strengthen their conclusions and the time consuming (I can imagine!) extra experiments / controls they have performed to convince me and the reader of the final message, i.e. the presence of receptor- and activity dependent distinct GPCR oligomeric state at the cell surface, dictating their spatial organization and, thus, potential ligand binding activity. I thus consider this manuscript as appropriate for publication with strong statistical & experimental work furthering our understanding of GPCR complexity.

I just have a very minor comment concerning typos in the introduction section: just after reference 5, you have two repetitions in the sentence ("will greatly depend" and "both the lipid composition of the plasma membrane").

ANSWER: we have now proceeded to this change in the manuscript

DETAILED ANSWERS TO REVIEWER 2

Reviewer #2 (Remarks to the Author):

The authors have made a lot of effort to address all of my comments to the original version of their manuscript. I feel that the manuscript has improved a lot both in terms of additional experiments that support the original conclusions of the manuscript as well as the clarity in writing.

Nevertheless, there are some remaining points that are puzzling and require further attention:

1) The manuscript still lacks a proper monomeric control. This can be either a GPCR, of course, but any other membrane protein that is strictly monomeric would be fine. This control is essential as it would prove that the method used by the authors is able to exclusively detect monomers as well. Moreover, this control experiment would rule out the possibility that this method is 'oligomeric-prone', i.e. that it tends to overestimate the amount of dimers or higher-order oligomers.

ANSWER 1: we totally agree with the reviewer that a monomeric control would have been fine. However, a proper monomeric control to validate our method is scientifically (and technically) impossible to obtain. This is why we decided to include in our previous revision a new synthetic dimeric control instead, as a more robust control (the β 2-AR chimera consisting in a fusion protein between 2 β 2-AR) for which the minimal distance (i.e. the monomeric chimera) depicted in AFM-SMFS experiments should be twice that of the β 2-AR and that we truly confirmed.

As a monomeric control, it would have been also interesting to test a dimerization-dead-mutant of our GPCRs (or others) or to test interference peptides to disrupt the dimerization interface. However, we don't know exactly the dimerization interface of GPCRs (which also changes when you express a receptor in different cell lines) and for the very few receptors for which it has been done, the receptor didn't anymore reach the cell surface thus preventing the AFM-SMFS studies. Testing another non-GPCR monomeric control (as it is usually done in TIRF studies, but rely on very old papers with still the assumption that these proteins do exist as exclusive monomers without being testing by other more recent techniques) could have been another option. However, the debate for the existence of monomers versus high-order complexes in GPCR field is the same for other single transmembrane proteins and more importantly, we don't know at all whether AFM-SMFS would be able to unfold one single transmembrane proteins with generally very complex extracellular domain, so it will be really difficult to compare with the GPCR results.

Indeed, nobody knows in biology whether any monomeric transmembrane protein does really exist as an exclusive monomer at the cell surface in living cells. This is truly a huge debate. To date, conclusions on the oligomerization state of a transmembrane protein rely on different independent studies using different receptor species, different cell lines, different receptor expression levels, different techniques. However, nobody has compared two different techniques in a similar cell system using similar receptor expression level. However, the expression level of the receptor is fundamental to take into consideration when studying GPCR oligomerization since it is well known (Calebiro et al. PNAS. 2013) that the expression level will directly impact the oligomeric populations with low receptor expression preferentially associated with monomeric population, while the increased expression of the receptor associated with high-

order receptor populations. Unfortunately, in TIRF studies (used by most of the recent studies on cell surface GPCR oligomerization) as well as in FRET-based studies (all fluorescent-based technologies), the authors cannot increase the expression levels of the receptor too much due to the technique, while, on the contrary, we cannot decrease too much the receptor expression level in our SMFS-studies otherwise we lose the detection sensitivity. Moreover, we used in our study transient transfection while most of the recent studies using TIRF or FRET have used instead stable-transfection which also changes a lot the experimental conditions and precludes any comparisons between these different studies.

In our revision, we have now, for the first time, compared two different techniques (Spectroscopy AFM and western-blot) in a similar cell system with similar receptor expression levels. **All western blot results obtained for the different GPCR tested perfectly fit with the results obtained with the AFM-SMFS analysis.** If the AFM-SMFS was not accurate to detect monomers and thus more prone to detect dimers/higher order oligomers, as stated by the reviewer 2, then we would have obtained similar GPCR profiles. By contrast, we found that the oligomeric profiles in AFM-SMFS (**and confirmed by the western-blot**) **are highly receptor-dependent** but also expression level-dependent (experiences with β 2-AR) as already known, which was somehow not surprising and is completely consistent with the previous TIRF studies (Calebiro et al. PNAS. 2013).

Further supporting the idea that GPCR oligomerization is highly receptor dependent, as you have suggested in the revision process, we also tested in addition to the μ OR, the **β 1-AR**, described as a high monomeric population in TIRF studies in classical CHO cells (*See results below*). Contrary to the HA- μ OR presented in the manuscript or the HA-mGlu3-R (**A, line 1**), in western-blot, in WTT CHO cells, we didn't detect any **β 1-AR** monomers but only high-order oligomers of the receptor (**A, line 3**). **Again, agreeing with these western-blot results, exactly same results** were obtained by the AFM-SMFS technique (**B**) with detection of dimers and higher-order oligomers of the HA- β 1-AR receptor only in these experimental conditions. Since these data didn't provide further information for the understanding and taking into account length restriction, we did not include these analyses in the revision manuscript.

A

I would like also to mention here that, initially, our paper was not to raise a debate about the existence of monomers, dimers etc. but more to validate the AFM-SMFS technique to unfold GPCRs at the cell surface in living mammalian cells. The different GPCRs, controls and techniques (AFM **AND** coimmunoprecipitation/western-blot) we have now included in our revised manuscript **have all confirmed** and strengthened the AFM-SMFS results and validated that the method is able to unfold **ALL the different oligomeric states of GPCRs**. As shown in our last revision, **the AFM-SMFS method can detect as many monomers of the new control (HA-β₂-AR x 2= HA-β₂-AR chimera) than the dimeric or high order oligomer populations**, but it also detects monomer populations for HA-β₂-AR, HA-mGlu3-R (See your point 3 below) or HA-AT1-R/AngII. Thus, the AFM-SMFS method is neither better prone to detect dimers/higher order oligomers than monomers but can detect as well GPCR monomers than dimers in our experimental conditions in a specific WTT-CHO cell line.

2) The new results on the μ-OR are interesting, however, it should be discussed more extensively why the authors' results are so different from what other labs have observed. Along this line it should be considered that μ-ORs have been reported to be essentially monomeric not only using single-molecule densities, but also by molecular brightness techniques (and FRET acceptor photobleaching) that are typically performed at much higher expression levels. As WTT-CHO cells and CHO-K1 cells are similar, this cannot be the only reason for the highly divergent results.

ANSWER 2: We would like first to highlight that the *WTT-CHO* cell line that we used in our study are not at all CHO-K1 or CHO-DG44 common cell lines, but has been specifically established to grow in suspension as well as on plastic (**see reference 6 of our manuscript**). This cell line is an opportunity for the SMFS-AFM technique that usually faces the problem of high non-specific adherence of the AFM-tip to the cell surface of most common mammalian cell lines. Thus, the cell system is also the originality of our AFM-SMFS study, otherwise similar AFM-SMFS experiments would have been done long before by the people from the AFM-field.

Now, we agree that our results with μ-OR are interesting and different from more recent studies using TIRF that identified this receptor as an **exclusive** monomer. By contrast, our

technique is also able to detect μ -OR as a monomeric population in mixture with high-order oligomers but **in perfect agreement with the results obtained in western-blot experiments performed in exactly** similar experimental conditions. Again, as mentioned in our answer 1, several experimental conditions can explain such discrepancies in between studies such as the use of different receptor species, different cell lines, different receptor expression levels, different transfection methods (transient-vs stable) and different techniques. However, nobody has compared two different techniques in a similar cell system using similar receptor expression level.

Moreover and importantly, recent studies on GPCR oligomerization using single-molecule imaging techniques (TIRF, FRET-based) do not rely on the use of native receptors but on GPCRs that have been fused to large tags (SNAP tag-182 amino acids~20 kDa or GFP-based tags-240 amino acids~27 kDa) at their N- or C-terminus that could interfere with oligomer complex formation and could also affect the diffusion rates in between different receptor populations. By contrast, in our study using AFM-SMFS, we used GPCRs fused at their N-terminus with an HA-Tag consisting in a 9 amino acid sequence (1 kDa) that didn't interfere with receptor expression and activity at all. Thus, divergent results concerning the oligomerization state of a GPCR between independent studies could also be attributed to these different experimental conditions. It is therefore crucial in the future to better understand the oligomerization state of a GPCR (or other transmembrane proteins) but by comparing different techniques using in a similar receptor-expressing cell line. This discussion about μ -OR results has been now introduced in the discussion section of the manuscript as proposed by the reviewer.

3) The mGluR3 is a Class C GPCR and an obligate dimer (or higher-order oligomer). How do the authors explain and justify that they observe a large population of monomeric receptors (39%, Figure 4b) in their measurements?

ANSWER 3: Our results with mGlu3-R receptor completely argue that the AFM-SMFS method is not more prone to detect more dimers or high-order GPCR oligomers since we detected high proportion (39 %) of mGlu3R monomers, as many as dimers (40 %). Again, these results are perfectly corroborated by the western-blot data in similar experimental conditions (**Extended data figure 7b**).

Now, results on the mGlu3-R have been previously and extensively discussed in the initial discussion section of our manuscript (*Second paragraph*). Moreover, despite it was long believed that mGluR receptor family does exist as obligatory dimers, more recent high-resolution techniques described for instance inhomogeneous distribution of mGlu4R in mouse cerebellum (native tissue) with multiple nanodomains with one to two mGlu4-R subunits but also higher order populations (Siddig et al, *Science Advances* 2020). Again, most results on mGluR receptors rely essentially on FRET-based techniques or TIRF studies and face the problems mentioned in our previous answer 2. Now, dimerization of mGluR receptors is a posttranslational process that relies on the establishment of disulfide bonds in the extracellular part of two monomers by specific enzymes. It is thus possible that some protomers of mGluR did not bind these enzymes and exist as monomers as seen in the recent study of Siddig et al in native tissues. Again, expression levels of the receptor, the use of specific cells, of native receptor can lead to different results for a similar receptor.

REVIEWERS' COMMENTS:

Reviewer #2 (Remarks to the Author):

I thank the authors for having responded to my previous comments in such exceptional detail and superb clarity. The revised manuscript now includes an absolutely fantastic discussion that incorporates all lines of evidence on the oligomerization state of GPCRs. Although intended to "validate the AFM-SMFS technique to unfold GPCRs at the cell surface in living mammalian cells", I truly believe that this manuscript in its current form is of much more substantial importance for the entire GPCR field, especially because of the much broader discussion of whether GPCRs exist as monomers, or dimers, or higher-order oligomers. I would totally agree with the authors that the issue of GPCR oligomerization is not even remotely understood (after all those years!) and that much more studies are required that meticulously compare different methods at the exact same conditions in different labs (most importantly fluorescence-based methods with AFM techniques). Thus, I am very much looking forward to seeing this manuscript published and to following the ensuing discussion in the field.